# Data-Driven Network Neuroscience: On Data Collection and Benchmark

**Jiaxing Xu**[1]*, **Yunhan Yang**[2]*, **David Tse Jung Huang**[2]*, **Sophi Shilpa Gururajapathy**[1]*,
**Yiping Ke**[1], **Miao Qiao**[2], **Alan Wang**[3,4,5], **Haribalan Kumar**[6,7], **Josh McGeown**[7]†, **Eryn Kwon**[3,7]†
partially for the Alzheimer's Disease Neuroimaging Initiative[‡]
[1]School of Computer Science and Engineering, Nanyang Technological University, Singapore
[2]School of Computer Science, The University of Auckland, New Zealand
[3]Auckland Bioengineering Institute, The University of Auckland, New Zealand
[4]Faculty of Medical and Health Sciences, The University of Auckland, New Zealand
[5]Centre for Brain Research, The University of Auckland, New Zealand
[6]General Electric Healthcare Magnetic Resonance, Australia & New Zealand
[7]Mātai Medical Research Institute, Tairāwhiti-Gisborne, New Zealand
`{jiaxing003, sophi.sg, ypke}@ntu.edu.sg`; `{h.kumar, j.mcgeown}@matai.org.nz`
`{yunhan.yang,dtj.huang, miao.qiao, alan.wang, e.kwon}@auckland.ac.nz`

## Abstract

This paper presents a comprehensive and quality collection of functional human brain *network* data for potential research in the intersection of neuroscience, machine learning, and graph analytics. Anatomical and functional MRI images have been used to understand the functional connectivity of the human brain and are particularly important in identifying underlying neurodegenerative conditions such as Alzheimer's, Parkinson's, and Autism. Recently, the study of the brain in the form of brain networks using machine learning and graph analytics has become increasingly popular, especially to predict the early onset of these conditions. A brain network, represented as a graph, retains rich structural and positional information that traditional examination methods are unable to capture. However, the lack of publicly accessible brain network data prevents researchers from data-driven explorations. One of the main difficulties lies in the complicated domain-specific preprocessing steps and the exhaustive computation required to convert the data from MRI images into brain networks. We bridge this gap by collecting a large amount of MRI images from public databases and a private source, working with domain experts to make sensible design choices, and preprocessing the MRI images to produce a collection of brain network datasets. The datasets originate from 6 different sources, cover 4 brain conditions, and consist of a total of 2,702 subjects. We test our graph datasets on 12 machine learning models to provide baselines and validate the data quality on a recent graph analysis model. To lower the barrier to entry and promote the research in this interdisciplinary field, we release our brain network data and complete preprocessing details including codes at `https://doi.org/10.17608/k6.auckland.21397377` and `https://github.com/brainnetuoa/data_driven_network_neuroscience`.

---

*Equal contribution

†Contribution was made exclusively to the data collection and community engagement of the Mātai dataset.

‡Part of the data used in preparation of this article were obtained from the Alzheimer's Disease Neuroimaging Initiative (ADNI) database (adni.loni.usc.edu). As such, the investigators within the ADNI contributed to the design and implementation of ADNI and/or provided data but did not participate in the analysis or writing of this report. A complete listing of ADNI investigators can be found at: `http://adni.loni.usc.edu/wp-content/uploads/how_to_apply/ADNI_Acknowledgement_List.pdf`

37th Conference on Neural Information Processing Systems (NeurIPS 2023) Track on Datasets and Benchmarks.

# 1 Introduction

Neuroscience unveils the principles and mechanisms underlying complex brain functions on normal subjects and subjects with neurological, psychiatric, and/or neurodevelopmental conditions. Recent years have witnessed an expansion in the size, scope and complexity of human neural data acquisition. Such an expansion joined with a rapid progress in machine learning and graph analytics has drawn a growing attention in network neuroscience – the neuroscience that comprehends the structure/function of the human brain using graphs (called brain networks) [14]. Each neuron/brain region can be modelled as a node and the interconnectivity between two neurons/brain regions as an edge. Brain networks capture rich structural and functional information that traditional manual examinations fail to capture: communications associated with spontaneous or task-evoked brain activities, dynamic patterns of neural signalling, interactions/disconnections associated with certain diseases, etc. [7].

Network neuroscience leverages on graph-based machine learning for clinically important applications. For example, a community in brain networks could represent areas of co-activation that could be weakened on brains with neurodegenerative conditions [34]. Graph classification [19] can help differentiate subjects with neurological diseases from healthy ones. Graph ordinal regression [38] can identify subjects with different stages of neurological diseases based on the severity. Nonetheless, *the potential of graph-based machine learning in clinical applications is hindered by the scarcity of available brain network datasets in this important interdisciplinary field.*

In this study, we focus on functional magnetic resonance imaging (fMRI), which monitors changes in the blood flow, i.e., blood-oxygen-level-dependent (BOLD) signals to capture functional activities [24]. The conversion of fMRI scans to brain networks has two stages, preprocessing and parcellation, both requiring intensive domain inputs. Specifically, to ensure the image quality for subsequent tasks, the preprocessing of raw MRI images needs proper quality control over a number of steps, including motion correction, realigning, field unwarping, normalization, bias field correction, and brain extraction. Different preprocessing choices lead to a large variation in the output images. Parcellation translates the preprocessed MRI images to regions-of-interest (ROIs) as nodes and the co-activation between ROIs as weighted edges. Each generated brain network contains i) a weighted adjacency matrix that characterizes the connectivity between ROIs and ii) a feature matrix that captures the attributes of ROIs in terms of the aggregated BOLD signals. Choosing a different brain atlas/scheme for parcellation leads to a different brain network. Existing study typically selects a single parcellation scheme to generate a fixed set of nodes for all subjects in the study, while the effect of different choices of group-wise data-driven parcellation schemes remains largely unexplored.

The above conversion poses a high barrier to entry the research on brain networks, limiting the development of data-driven network neuroscience. Specifically, domain knowledge in neuroimage preprocessing is required to select and guide the proper pipeline and tools used, image processing and graph extraction lead to high computational costs, and large-scale imaging studies require a complex setup with multi-site, multi-scanner, and multiple acquisition protocols. This paper aims to bridge the gap by making more brain network data available to the public. We believe that releasing this brain network collection will promote research in the interdisciplinary field of network neuroscience, machine learning, and graph analytics, and advance graph-based and clinical studies such as the detection of neurodegenerative conditions. Our main contributions are highlighted below.

1. We release a large resting-state functional brain network collection to the public. The collection was originated from 6 raw rs-fMRI image sources with 5 well recognized ones in neuroscience and one new data source, covering 3 neurodegenerative and one brain injury conditions, i.e., Autism, Alzheimer, Parkinson, and mTBI. The collection consists of ABIDE (N=1025), ADNI (N=1327), PPMI (N=209), Mātai (N=60) and two other sources totalling to 2,702 subjects.

2. We test the datasets on one recent graph analysis model [34], 6 conventional machine learning (ML) models, as well as 6 representative graph ML models. The experimental results demonstrate that the quality of our datasets is not compromised by the conversion process and can serve as a domain benchmark for subsequent research.

# 2 Related Work

**Data Collection.** Preprocessed Connectomes Project (PCP) [17] is an initiative to preprocess part of the raw MRI images in the International Neuroimaging Data-sharing Initiative (INDI) database

Table 1: Statistics of our datasets and the generated resting-state functional brain networks. Each subject has a graph (brain connectivity network) generated under each Parcellation Method (PM) of AAL, HarvardOxford (HO), Schaefer, k-means and Ward Clustering (see Table 2 for details). The number of nodes in a graph generated under a PM is the number of ROIs of the PM. We call an edge non-zero if its weight has absolute value $> 10^{-2}$. The number of non-zero edges varies under different parcellations. The number of node features is the length of the BOLD signals.

| Dataset | Condition | # of Graphs (# of Subjects) | # of Classes | Avg # Non-Zero Edges under PM (# of Nodes) | | | | | Avg # of Node Features |
| | | | | AAL (116) | HO (48) | Schaefer (100) | $k$-means (100) | Ward (100) | |
|---|---|---|---|---|---|---|---|---|---|
| ABIDE | Autism | 1025 | 2 | 6402 | 1112 | 4811 | 4698 | 4729 | 201 |
| ADNI | Alzheimer | 1327 | 6 | 6447 | 1112 | 4824 | 4734 | 4715 | 344 |
| PPMI | Parkinson | 209 | 4 | 6512 | 1122 | 4866 | 4795 | 4684 | 198 |
| Mātai | mTBI | 60 | 2 | 6433 | 1112 | 4832 | 4750 | 4731 | 198 |
| TaoWu | Parkinson | 40 | 2 | 6481 | 1116 | 4846 | 4724 | 4766 | 239 |
| Neurocon | Parkinson | 41 | 2 | 6455 | 1114 | 4830 | 4677 | 4779 | 137 |

and make *the preprocessed neuroimages* publicly available. Within PCP, one relevant dataset that has gone through functional preprocessing pipelines is the *ABIDE dataset* on Autism. Note that the pipeline used for preprocessing ABIDE was proposed in 2012 [9] while the state-of-the-art functional preprocessing pipeline is fMRIPrep [20] which introduces less uncontrolled spatial smoothness [20] compared to other preprocessing tools. Some work converts preprocessed neuroimages to brain network datasets which, however, are predominately binarized, i.e., the edge weight can take only two values 0 or 1. For example, in [34], the ABIDE dataset was converted to binarized brain networks. In [42, 1], around 80 samples for Attention Deficit Hyperactivity Disorder (ADHD) were released in three datasets KKI, OHSU, and Peking_1 in the form of binarized brain networks. *There still lacks a large collection of quality brain network datasets available to the public.* Our collection uses fMRIPrep on data from 6 sources (see Section 3 for details) on 4 clinical conditions of interest under different parcellation schemes and wraps the whole conversion process from raw MRI images to brain networks in a holistic manner. We release the codes of the entire processing pipeline and keep future efforts in refining the pipeline and/or enriching the collection open.

**Benchmark.** Network neuroscience that uses machine learning and graph analytics has attracted an increasing attention [23]. Along this line of research, most recent studies [10, 46, 37, 30, 34] apply machine learning models to perform connectivity analysis on the ABIDE dataset with the generated brain networks. In our benchmark, we tested our datasets on one recent graph analysis model [34], 6 conventional ML models and 6 representative graph ML models: the quality of our datasets is not compromised by the conversion process and can serve as baselines for this line of research.

## 3 Dataset Sources: Raw Neuroimages

This section describes our selected sources of raw neuroimages and their selection and acquisition settings. Table 1 summarizes our datasets and the generated brain networks. This paper focuses on resting-state functional connectivity of the brain using rs-fMRI, a potent method for detecting neurodegenerative conditions [24], leaving alternative modalities for future exploration. To achieve quality image preprocessing, each rs-fMRI image needs a structural T1-weighted (T1w) image that was acquired from the same subject in the same scan session. T1w image provides structural details which allow brain mask extraction, image alignment and BOLD time series normalization. Other pipelines such as DPARSF [54] have the same requirement.

**Autism Brain Imaging Data Exchange (ABIDE)** The ABIDE initiative aggregated functional brain imaging data collected from laboratories around the world to support the research on Autism Spectrum Disorder (ASD). ASD has stereotyped behaviors such as irritability, hyperactivity, depression, and anxiety. Subjects are classified into typical controls and those suffering from ASD.

**Alzheimer's Disease Neuroimaging Initiative (ADNI)** ADNI [4, 3, 52] is a longitudinal multisite study for the early detection and tracking of Alzheimer's Disease (AD). AD is a progressive neurologic disorder that causes the brain to shrink and brain cells to die and is the most common cause of dementia that affects a person's ability to function independently. ADNI data used in this study were obtained from the ADNI database (adni.loni.usc.edu) which was launched in 2003 as a public-private partnership, led by Principal Investigator Michael W. Weiner, MD. The primary goal of ADNI has

been to test whether serial MRI, positron emission tomography (PET), other biological markers, and clinical and neuropsychological assessment can be combined to measure the progression of mild cognitive impairment (MCI) and early Alzheimer's disease (AD). Subjects are from 6 different stages of AD: cognitive normal (CN), significant memory concern (SMC), mild cognitive impairment (MCI), early MCI (EMCI), late MCI (LMCI), and Alzheimer's disease (AD) [8].

**Parkinson's Progression Markers Initiative (PPMI)** PPMI aims to identify biological markers of Parkinson's risk, onset and progression. Parkinson's disease is a progressive nervous system disorder that mainly affects movement [41]. The study is ongoing and contains multimodal, multi-site MRI images similar to ADNI. The PPMI dataset contains subjects from 4 classes: normal control, scans without evidence of dopaminergic deficit (SWEDD), prodromal, and Parkinson's disease (PD).

**Mātai** Mātai is a longitudinal single site, single scanner study designed for detecting subtle changes in the brain due to a season of playing contact sports. This new dataset consists of the brain networks preprocessed from the data collected from Gisborne-Tairāwhiti area, New Zealand, with 35 contact sport players imaged at pre-season (N=35) and post-season (N=25) with subtle brain changes confirmed using diffusion imaging study due to playing contact sports. Note that this dataset does not release raw data nor metadata and has been preprocessed so that no ancestral history is able to be extracted.

**TaoWu and Neurocon** TaoWu and Neurocon datasets are released by ICI [6] and are two of the earliest image datasets released for Parkinson's. The datasets consist of age-matched subjects captured using a single machine and on a single site. We include these two datasets in our collection as they could be used in studies that aim to minimize or contrast the variability introduced from different image acquisition settings. It includes normal controls and patients labelled with PD. Neurocon and Taowu label patients with a diagnosis of PD who have been under treatment (most under levodopa) as PD. PPMI's PD definition involves patients with a diagnosis of PD for two years or less and who are not taking PD medications. Under these definitions, Neurocon and Taowu are more similar when compared to PPMI. It is worth noting that while these two have similar scanning protocols, they used different scanners (with Taowu being higher in resolution). In [6], the authors compared these scans and argued that they can be treated similarly. We believe there could be more such explorations with the data available, which is one of the main reasons why we want to release this collection.

## 4 From MRI Images to Brain Networks: Design Choices

We adopt the common functional processing pipeline in network neuroscience [21] to convert raw MRI images (rs-fMRI and T1w) into brain networks, as depicted in Figure 1. Under this general pipeline, a number of choices need to be made in data selection, data formats, neuro-image preprocessing tools, parcellation schemes, network edge formation, etc. We worked closely with our domain experts to ensure that our design choices are sensible and state-of-the-art. Specifically, Step A collects raw MRI images based on the selection criteria (Section 4.1). Step B converts the images into BIDS format (Section 4.2). Step C preprocesses the images using fMRIPrep (Section 4.3). Step D parcellates the preprocessed data into different ROIs (Section 4.4). Steps E-F extract the connectivity matrix and the feature matrix and form the brain network (Section 4.5). Domain experts have guided our preprocessing by providing advice/feedback on i) the selection of images from the data sources, ii) the choices of using the state-of-the-art fMRIPrep pipeline, iii) parcellation strategies, iv) quality check of fMRIPrep outputs, and v) the selection of confounds.

### 4.1 Data Collection and Selection Criteria

Our data sources involve multi-site, muti-scanner images. Thus, the inclusion criteria of our data collection is based on the availability, validity, and quality of the MRI images of our required types.

For ABIDE, TaoWu, and Neurocon, the original source images were already collated with 1 raw rs-fMRI image paired with 1 T1w image for each subject. We included all subjects provided by these data sources except for those that had quality issues. A subject has quality issues if it has an incomplete image (*i.e.*, not containing the full brain) and/or damaged data (*i.e.*, with error reported by any subsequent preprocessing steps). Available metadata of each subject from the source is included in our collection, such as age and gender. The gender distributions of our datasets closely match

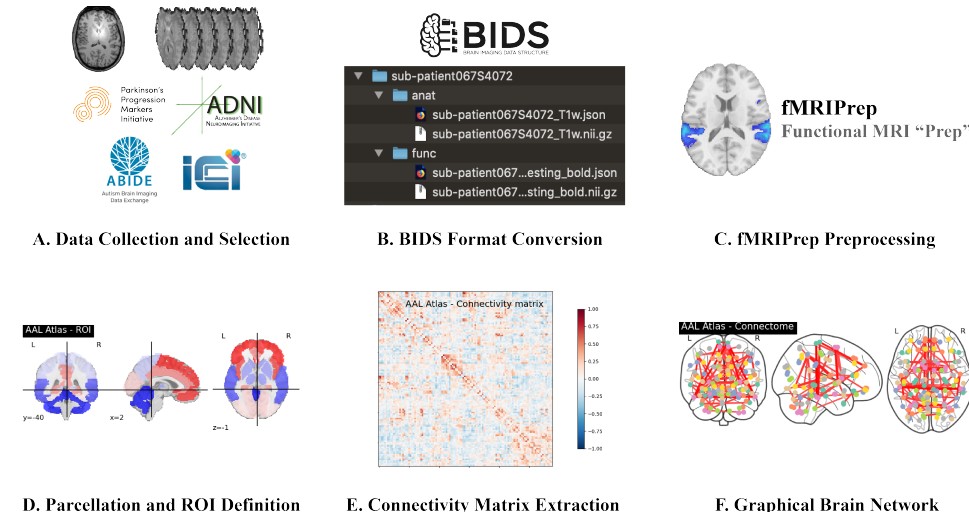



A. Data Collection and Selection     B. BIDS Format Conversion     C. fMRIPrep Preprocessing

D. Parcellation and ROI Definition     E. Connectivity Matrix Extraction     F. Graphical Brain Network



Figure 1: Brain Network Construction Pipeline

those of the original data sources. The only exception is ADNI: our released data contains 0.6% more females than the original ADNI data.

For longitudinal study data sources ADNI, PPMI, and Mātai, it is possible that some subjects have had multiple scans over the course of several years. Depending on the scanning protocol for the study, different types of images were taken at various times (*e.g.,* baseline, 1 year follow-up, 2 year follow-up, etc.). The baseline study (the first scan) is usually the most comprehensive one that would cover a wide range of modalities. Thus, as suggested by our domain experts, we consider the baseline study which is likely to be the set of scans with both an rs-fMRI image and a T1w image taken on the same date. In the case that multiple rs-fMRI and T1w scans exist, we selected the first available one.

ABIDE, ADNI, and PPMI are multi-site neuroimaging sources. We chose not to apply data harmonization techniques to these datasets in our preprocessing due to the following reasons. (1) For fMRI connectivity, there is no well-grounded harmonization method for benchmark. (2) Even though no fMRI connectivity harmonization was applied, [43] has shown that fMRI connectivity still has some good repeatability. (3) The site and scanner information are available and researchers have the flexibility of performing harmonization based on our data collection for further analysis.

For validity, we examined each image manually to make sure the images conformed to their labels in the database and did not have obvious data format issues for preprocessing. We had to check this manually because the image databases often had inconsistent labeling.

## 4.2 BIDS Conversion

Raw MRI images are typically in the Digital Imaging and Communications in Medicine (DICOM) format. DICOM images are then converted to the Neuroimaging Informatics Technology Initiative (NIfTI) format using dcm2niix [36], which includes a JSON file that details various imaging parameters such as scanner model, magnetic field strength, flip angle, slice timing, echo time and repetition time. The NIfTI and JSON files are then organized into a folder hierarchy with a precise naming convention known as Brain Imaging Data Structure (BIDS) [27]. After conversion, BIDS formatted data can be applied to preprocessing pipelines in a highly reproducible and transparent manner.

## 4.3 fMRIPrep Preprocessing

BIDS compliant datasets are then preprocessed using fMRIPrep [20], a state-of-the-art tool for preprocessing fMRI. fMRIPrep performs basic processing steps (coregistration, normalization, noise component extraction, skull stripping, etc.) and uses a combination of tools from well-known software packages, including FMRIB Software Library (FSL) [31], Analysis of Functional NeuroImages (AFNI) [16], Advanced Normalization Tools (ANT) [5] and FreeSurfer [22]. This pipeline was designed to use the best software implementation for each step of preprocessing. fMRIPrep can be

Table 2: Parcellation Methods

| Name | # ROIs | Generation Method |
|---|---|---|
| AAL [49] | 116 | Delineated with respect to anatomical landmarks by following the sulci course in the brain. |
| HarvardOxford (HO) [40] | 48 | Created by subdividing neocortex by topographic criteria into 48 parcellation units corresponding to the principal cerebral gyri. |
| Schaefer [47] | 100 | Using gradient-weighted Markov Random Fields (gwMRF) to automatically group similar fMRI regions. |
| $k$-means Clustering [39, 2] | 100 | Top-down clustering algorithm that partitions voxels into non-overlapping predefined number of regions. |
| Ward Clustering [51, 2] | 100 | Bottom-up hierarchical clustering algorithm that agglomerates together voxels progressively into regions. |

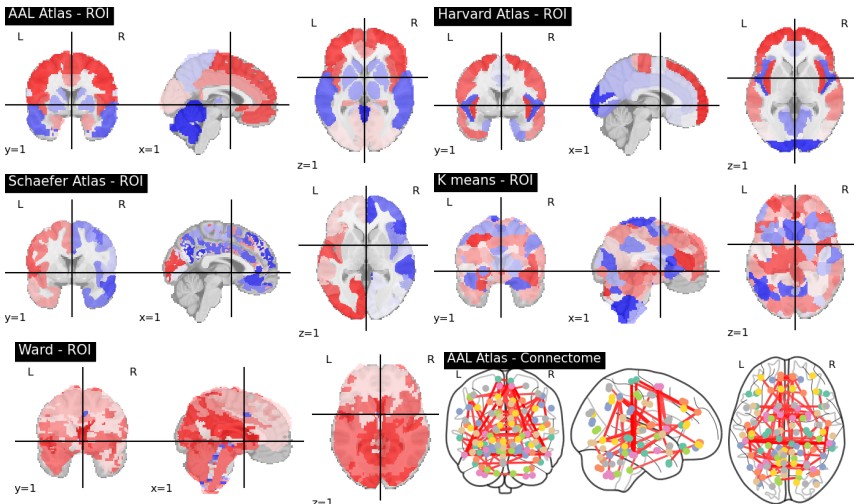

Figure 2: Extracted ROIs and Glass Brain Connectome (right bottom corner)

installed via container technologies such as docker or singularity. fMRIPrep automates the pipeline by using the scanner outputs from the images (*e.g.*, slice timing correction parameters are taken automatically from the scanner outputs of the original image data). Therefore, the user does not need to specify these parameters manually. fMRIPrep can be configured to include or exclude specific workflow steps, such as ignoring slice timing. We followed the default automatic fMRIPrep workflow with surface reconstruction enabled. No further modifications to the settings were made as the default workflow met our expectations for a functional preprocessing pipeline. As part of our validation process of fMRIPrep, we have passed the preprocessed images over to our MRI imaging experts: they have carefully examined the outputs to ensure the data quality. fMRIPrep outputs conform to the BIDS Derivatives specification for compatibility and include the preprocessed BOLD images and the confounds file, which records fluctuations during MRI data acquisition, also known as nuisance regressors. fMRIPrep is computationally expensive. In our case, one run on one subject of fMRIPrep using 8 threads on an Intel(R) Core(TM) i9-10940X CPU@3.30GHz took on average 4-5 hours.

### 4.4 Parcellation Strategies

Parcellation defines regions in the brain known as region-of-interest (ROI). The parcellation step takes in the two outputs from fMRIPrep for each subject to generate the parcellations: the preprocessed BOLD image and the confounds file. We adopted the standard 9 parameters (9P) confounds setting widely used in functional connectivity studies [25, 26] for denoising: white matter, cerebrospinal fluid, global signal, and the 6 rigid-body motion parameters for rotation and translations at $x$, $y$ and $z$ axes. To parcellate the brain, we selected a list (Table 2) of Parcellation Methods (PMs) that cover both atlas-based and clustering-based PMs frequently used in the domain. Brain regions partitioned through predefined regions are atlas-based PMs. We selected the most commonly used atlas-based PMs, AAL, HarvardOxford (HO), and Schaefer, that are generated using different strategies and

brain features. Clustering-based parcellations partition the brain by computing the similarity in the BOLD signals between voxels and performing clustering on the voxels based on the similarities. Each subject directly generates its own set of ROIs. $k$-means and Ward clustering are the two methods used in the domain. We included them and set the number of ROIs as 100, which is close to the predefined value in atlas-based PMs. Our datasheet in the Supplementary details the file format and output files. Figure 2 shows examples of the ROIs extracted by these methods.

### 4.5 Brain Network Extraction

The BOLD signals from parcellated ROIs are used to compute the connectivity matrix of the brain network. The matrix captures functional relationships between parcellated regions and reflects their co-activations. Connectivity matrix is extracted using a built-in function of nilearn [2] called ConnectivityMeasures [55]. It takes 2 inputs, BOLD signals of the brain regions and the connectivity metric (*e.g.*, correlation). The former is obtained by averaging the BOLD signals of all the voxels lying in the same ROI. Figure 1E shows an example of a connectivity matrix. Among existing connectivity measures (correlation, covariance, partial correlation, etc.), correlation is used by a majority of graph-based neuroscience research [34, 37, 44]. Thus, in our released collection, we select correlation as the connectivity measure and make the fully weighted matrices available without performing any thresholding on the correlation values. For the practitioners who are interested in alternative measures, we have uploaded preprocessed images (excluding the Mātai dataset) and our matrix generation code as part of our collection, which enables the researchers to generate their own matrices based on their desired measure. The datasheet in our Supplementary details our outputs and file formats. The connectivity matrix can be represented equivalently as a glass brain connectome. An example connectome plot on the AAL atlas is shown in the right bottom corner of Figure 2. The connectome plot shows top 1% of brain functional connectivities for visualization.

The brain network constructed by this pipeline has the set of ROIs as nodes and their co-activations (measured as Correlation) as edges. Each brain network is represented by a weighted adjacency matrix and a feature matrix. The former is the connectivity matrix extracted and the latter stores the BOLD signals of ROIs. Note that we include the BOLD signals as node feature matrices in our data release for data completeness: some analytical methods on brain networks may want to utilize such information directly or further extract features from them.

## 5 Data Quality Assessment and Baseline Comparisons

To assess if the graph representation maintains the data quality from neuroimages, we test our brain network data collection on two tasks, graph classification and graph ordinal regression. The aim of graph classification is to distinguish patients from healthy subjects for datasets with two classes (ABIDE, Mātai, TaoWu, and Neurocon); and to distinguish subjects among different disease stages and healthy groups by treating each class independently when it comes to multi-class datasets (ADNI and PPMI). In contrast, graph ordinal regression is only applied to multi-class datasets by differentiating stages based on the disease severity.

For classification, we test on both conventional ML models typically used in neuroscience and representative graph ML models. For conventional models, we follow the common practice in the domain [10, 30] to vectorize the input connectivity matrices by flattening them. For ordinal regression, we test on the classic logistic ordinal regression model. To further validate the quality of our data, we also test on a recent graph analysis model on functional networks for classification [34]. Finally, we also include a sensitivity study on the number of ROIs and the training set size. All these models only utilize the connectivity matrices for assessing the quality of our brain network construction. The data is split to 8:1:1 for training, validation, and testing with 10-fold cross-validation performed.

### 5.1 Results on Classification

We select 6 conventional machine learning models for this study: Logistic Regression (LR), Gaussian Naive Bayes (NB), Support Vector Machine Classifier (SVC), $k$-Nearest Neighbours (kNN), Random Forest (RF), and Multi-Layer Perceptron (MLP) [29]. We used the implementation from the scikit-learn library [45, 11] with Grid Search for model selection. We also select 6 typical graph-based machine learning models, including GCN [33], GraphSAGE [28], GIN [53], GAT [50], GatedGCN

Table 3: Classification accuracy (mean±standard deviation) on conventional ML methods. The best result at each parcellation is highlighted in bold. The best result in each dataset is underlined.

| | **ABIDE** | | | | | **ADNI** | | | | |
|---|---|---|---|---|---|---|---|---|---|---|
| | AAL | HO | Schaefer | $k$-means | Ward | AAL | HO | Schaefer | $k$-means | Ward |
| LR | 63.8±3.0 | **63.6±4.2** | **64.8±3.7** | 48.4±4.4 | 51.1±5.8 | **64.1±1.8** | 61.9±2.1 | 62.0±4.2 | 58.6±2.6 | 60.4±0.8 |
| NB | 60.4±5.5 | 59.2±5.4 | 61.6±3.6 | 51.6±3.7 | **54.2±5.0** | 53.6±4.4 | 52.7±6.7 | 48.9±3.3 | 32.5±3.6 | 55.0±2.6 |
| SVC | **65.7±3.3** | 62.9±3.6 | 64.4±5.1 | 49.3±4.1 | 53.2±4.9 | 63.4±1.9 | **66.2±2.9** | 61.5±5.0 | **61.8±0.3** | **61.8±0.3** |
| kNN | 58.1±5.3 | 56.1±4.5 | 59.7±3.5 | 50.7±2.9 | 49.3±2.3 | 60.6±2.1 | 62.9±3.8 | **63.1±2.4** | 31.4±22.9 | 59.5±3.9 |
| RF | 62.4±2.8 | 60.5±3.1 | 62.6±2.6 | 49.5±5.4 | 51.3±2.9 | 61.9±2.1 | 61.7±2.8 | 62.1±1.8 | 61.5±0.4 | 61.6±0.5 |
| MLP | 54.4±1.2 | 62.2±4.4 | 49.6±4.4 | **63.6±3.9** | 51.7±6.3 | 62.4±1.8 | 62.9±2.2 | 46.7±5.3 | 61.7±0.2 | 48.6±5.9 |
| | **PPMI** | | | | | **Mātai** | | | | |
| | AAL | HO | Schaefer | $k$-means | Ward | AAL | HO | Schaefer | $k$-means | Ward |
| LR | 56.0±7.8 | 56.0±9.2 | 56.5±6.8 | 60.3±7.4 | 60.3±8.8 | 66.7±21.1 | **58.3±13.4** | 60.0±20.0 | 55.0±18.3 | **58.3±20.1** |
| NB | 58.4±5.2 | 52.6±8.6 | 57.5±7.6 | 58.4±5.7 | **62.2±7.4** | **68.3±22.9** | 45.0±21.2 | 56.7±18.6 | 55.0±15.0 | 53.3±18.0 |
| SVC | **64.1±5.7** | **63.6±6.4** | **63.2±8.6** | 60.8±7.5 | 60.8±8.9 | 65.0±20.3 | **58.3±13.4** | 56.7±17.0 | 58.3±17.1 | **58.3±17.1** |
| kNN | 51.2±9.1 | 55.5±7.3 | 53.5±7.6 | 60.3±6.6 | 60.8±8.9 | 61.7±16.8 | 50.0±21.1 | 58.3±18.7 | 58.3±17.1 | 48.3±13.8 |
| RF | 61.6±8.8 | 62.6±9.9 | 62.6±8.1 | 58.4±9.2 | 61.7±7.3 | 60.0±18.6 | 53.3±20.0 | **63.3±24.5** | 48.3±17.4 | 53.3±20.8 |
| MLP | 57.8±10.4 | 62.2±8.0 | 57.9±5.0 | 57.4±9.2 | 52.6±5.7 | 45.0±19.8 | 48.3±21.7 | 53.3±18.0 | **63.3±18.0** | 50.0±22.4 |
| | **TaoWu** | | | | | **Neurocon** | | | | |
| | AAL | HO | Schaefer | $k$-means | Ward | AAL | HO | Schaefer | $k$-means | Ward |
| LR | **77.5±17.5** | **72.5±23.6** | **75.0±15.0** | 50.0±15.8 | **52.5±17.5** | **68.5±25.1** | 61.5±26.3 | 65.5±24.9 | 58.0±22.9 | 63.0±25.9 |
| NB | 65.0±15.8 | 60.0±24.5 | 62.5±23.6 | 37.5±30.1 | 50.0±19.4 | 58.0±22.9 | 59.0±28.4 | 63.5±23.3 | 48.0±24.2 | 55.5±22.3 |
| SVC | 67.5±17.5 | 67.5±17.5 | 65.0±15.0 | 52.5±7.5 | 50.0±19.4 | 63.0±25.9 | **68.5±25.9** | **69.0±25.9** | 63.0±25.9 | 63.0±25.9 |
| kNN | 55.0±21.8 | 60.0±16.6 | 65.0±16.6 | 42.5±16.0 | 50.0±0.0 | 65.0±25.5 | 49.0±27.6 | 55.5±24.9 | 63.0±25.9 | 63.0±25.9 |
| RF | 65.0±11.3 | 60.0±22.9 | 57.5±25.1 | 40.0±32.0 | 47.5±20.8 | 55.5±24.9 | 61.0±29.8 | 58.5±22.4 | 58.0±22.9 | 63.0±25.9 |
| MLP | 60.0±20.0 | 67.5±16.0 | 42.5±22.5 | **57.5±27.5** | 45.0±21.8 | 61.0±11.8 | 67.5±16.0 | 63.5±11.8 | **63.5±11.8** | **63.5±11.8** |

Table 4: Classification accuracy (mean±standard deviation) on graph ML methods and BOLD time series based methods with Schaefer parcellation. The best result in each dataset is in bold, with those underlined indicating superior performance to conventional ML methods.

| | ABIDE | ADNI | PPMI | Mātai | TaoWu | Neurocon |
|---|---|---|---|---|---|---|
| GCN | 61.0±2.8 | 61.6±0.6 | 54.0±9.1 | 56.7±18.6 | 60.0±29.2 | 59.0±20.7 |
| GraphSAGE | 63.1±3.1 | 61.2±1.7 | 55.0±12.9 | 61.7±10.7 | 60.0±33.9 | 68.5±15.2 |
| GIN | 57.0±3.9 | 61.9±0.4 | **57.9±8.1** | 48.3±13.8 | 65.0±20.0 | 68.5±15.2 |
| GAT | 60.9±5.0 | 61.3±1.3 | 55.0±8.0 | **66.7±18.3** | **67.5±22.5** | 54.0±15.6 |
| GatedGCN | 63.6±4.7 | **62.1±4.5** | 52.6±11.5 | 58.3±8.3 | 65.0±22.9 | **69.0±25.5** |
| BrainNetCNN | **65.8±2.5** | 61.1±2.9 | 57.3±10.3 | 61.7±13.3 | 65.0±27.8 | 66.0±22.5 |
| GRU | 53.6±1.4 | 61.8±0.3 | 54.1±2.2 | 58.3±8.3 | 50.0±0.0 | 63.8±1.5 |
| 1D-CNN | 55.5±3.0 | 50.2±4.0 | 55.0±11.5 | 61.7±13.0 | 45.0±15.0 | 58.5±11.2 |

[12], and BrainNetCNN [32]. The implementations mainly follow those in [18]. To compare the effectiveness of brain networks with that of BOLD signals in disease classification, we test on two methods based on BOLD signals: GRU [15] and 1D-CNN [35].

Table 3 reports the classification accuracy of conventional ML methods on the datasets using different parcellation methods. In most cases, the overall accuracy falls within the range of 60% to 70%, which is consistent with the results reported in previous studies [34, 10, 46, 37, 30], despite some differences in data filtering and preprocessing. With respect to parcellation methods, we observe that atlas-based parcellation methods in general outperform clustering-based methods. The best performance on each dataset always occurs when AAL, HO or Schaefer is applied. Particularly, AAL achieves the best performance in 4 out of 6 datasets. The inferiority of clustering-based parcellation is likely due to the effects from some randomness in the clustering process, *e.g.*, initial centroid selection in $k$-means. Among different learning models, LR and SVC perform better in most cases. This also explains why they are commonly used as baseline models in the domain [34, 37].

Table 4 reports the classification results of graph-base ML methods and BOLD time series based methods on the datasets with the Schaefer parcellation. No single graph ML method can consistently dominate across datasets. In 3 out of 6 datasets, the best graph ML method outperforms the best of conventional ML methods, which shows the potential of graph-based models. Compared with both conventional and graph ML models, the model performance based on BOLD signals is significantly worse. This verifies the effectiveness/usefulness of brain networks in the task of disease classification.

Table 5: Accuracy (mean±standard deviation) on logistic ordinal regression (LOR) vs logistic regression (LR) on ADNI and PPMI. The best result at each parcellation is highlighted in bold.

| | ADNI | | | | | PPMI | | | | |
| | AAL | HO | Schaefer | $k$-means | Ward | AAL | HO | Schaefer | $k$-means | Ward |
|---|---|---|---|---|---|---|---|---|---|---|
| LR | 64.1±1.8 | 61.9±2.1 | 62.0±4.2 | **58.6±2.6** | **60.4±0.8** | **56.0±7.8** | 56.0±9.2 | **56.5±6.8** | **60.3±7.4** | 60.3±8.8 |
| LOR | **65.6±2.2** | **62.1±3.0** | **62.8±2.2** | 58.5±5.5 | 60.2±3.3 | 54.1±13.0 | **57.5±8.1** | 55.6±8.9 | 59.4±8.5 | **60.8±9.1** |

Table 6: Results of CS-P1 [34] on the ABIDE dataset used in [34] and on our ABIDE dataset

| | | ABIDE used in [34] | | | Our ABIDE | | |
| | | Accuracy in [34] | Reproduced Accuracy | | | Reproduced Accuracy | |
| Subgroup | # Graphs | Method 1 | Method 1 | Method 2 | # Graphs | Method 1 | Method 2 |
|---|---|---|---|---|---|---|---|
| Adolescents | 237 | 72.0±7.0 | 71.4±3.7 | 64.9±7.8 | 259 | 72.2±5.5 | 68.3± 3.1 |
| Children | 101 | 86.0±7.0 | 83.0±9.6 | 69.5±8.6 | 96 | 80.1±2.6 | 67.1±14.0 |
| Eyesclosed | 294 | 71.0±3.0 | 68.7±4.6 | 60.9±3.7 | 312 | 68.6±3.5 | 64.8± 6.2 |
| Male | 838 | 63.0±1.0 | 63.5±2.3 | 60.4±5.1 | 873 | 65.3±2.1 | 62.7± 4.0 |

The accuracy scores in Tables 3 and 4 might appear relatively modest, raising concerns about the clinical applicability of solving the classification problem. It's important to note that disease classification through brain networks is a nascent and evolving research domain. Many of the ML methods tested are general-purpose and not specifically designed for brain networks. Our release of fully weighted brain network matrices in this collection is anticipated to stimulate and advance research in this field. This initiative is expected to foster the development of tailored learning models for brain networks, yielding clinically relevant outcomes. The current results can serve as foundational benchmarks for future research endeavors.

## 5.2 Results on Ordinal Regression

With the two multi-class datasets ADNI and PPMI, we study the problem of ordinal regression, in which the classes are ordered by the severity of the corresponding disease. We select the classic logistic regression model for this epxeriment and use the implementation in the scikit-learn library. Table 5 reports the results of the logistic ordinal regression (LOR) versus the logistic regression (LR). The results show that LOR does not exhibit superiority to LR. One potential reason may be its sensitivity to the class imbalance problem: ADNI and PPMI have very skewed class distribution as shown in Table II in the supplementary. With the availability of the datasets, researchers could further inspect this issue and design better ordinal regression solutions to handle them.

## 5.3 Data Quality Study on ABIDE Dataset with a Graph Analysis Approach for Classification

To evaluate the quality of our data collection, we performed a recent graph-based functional analysis approach [34] to both an existing brain network dataset derived from ABIDE and our processed brain connectivity networks of the ABIDE for disease classification. The approach in [34] (model version of CS-P1) first computes two summary connectivity matrices $C$ and $P$, respectively from the Control group and the Patient group. It then extracts the dense contrast subgraphs from $C$ and $P$ and uses them as features for classification. Essentially, a dense contrast subgraph refers to a subset of nodes whose induced subgraph is dense in $C$ and sparse in $P$ (namely contrast), or vice versa. It can be considered an Optimal Quasi-Clique (OQC) problem [48] and solved by the DENSDP approach [13]. To make the classification fairer, we list two methods of finding the contrast dense subgraph for CS-P1 [34]. Method 1 uses the whole dataset (all the subjects in both training and test sets) to extract the contrast dense subgraph. The results in [34] used this method. Method 2 uses only the training set to extract the contrast dense subgraph.

We first applied the two methods of CS-P1 to the functional networks of different subgroups of the ABIDE dataset: both the code and data were provided by the authors of [34]. The results using Method 1 (left part of Table 4) are generally comparable with the results reported in [34]: the reproduced accuracy is 3% lower than the claimed accuracy in the subgroup of "Children", and 2% lower in the subgroup of "Eyesclosed"; the results on the other two subgroups are consistent. The accuracy results using Method 2 are much lower than the results using Method 1.

We then applied CS-P1 (the same code) on our functional networks of subgroups of ABIDE dataset. Table 6 (right) shows the result. Note that the subjects in different subgroups may overlap. Due

Table 7: Classification accuracy (mean±standard deviation) when tuning #ROIs in Schaefer parcellation. The best result in each method is in bold and the best result in each dataset is underlined.

| | ABIDE | | | | TaoWu | | | |
|---|---|---|---|---|---|---|---|---|
| | 100 | 200 | 500 | 1000 | 100 | 200 | 500 | 1000 |
| LR | 64.8±3.7 | **67.9±3.8** | 67.4 ±3.8 | 67.6±4.7 | **75.0±15.0** | 57.5±22.5 | 52.5±23.6 | 52.5±17.5 |
| NB | 61.6±3.6 | **62.4±3.6** | 61.9±3.0 | 60.5±3.5 | **62.5±23.6** | 62.5±23.1 | 60.0±20.0 | 60.0±20.0 |
| SVC | **64.4±5.1** | 63.7±4.4 | 62.4±3.8 | 63.1±4.0 | **65.0±15.0** | 57.5±22.5 | 55.0±24.5 | 50.0±19.4 |
| kNN | **59.7±3.5** | 59.5±5.1 | 58.0±3.8 | 55.5±5.8 | **65.0±16.6** | 52.5±23.6 | 50.0±25.0 | 50.0±25.0 |
| RF | **62.6±2.6** | 62.4±3.2 | 61.6±3.2 | 62.1±5.3 | **57.5±25.1** | 55.0±26.9 | 52.5±26.1 | 45.0±18.7 |

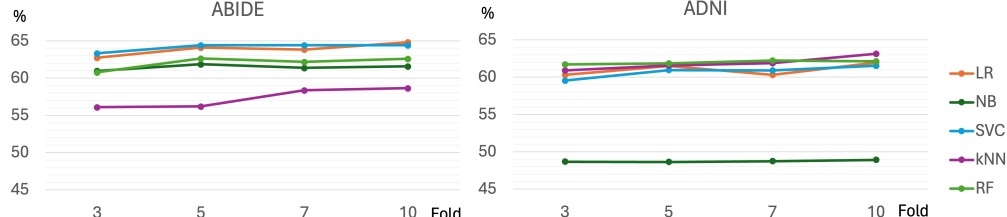

Figure 3: Test accuracy on ABIDE and ADNI with Schaefer when tuning training set size.

to different data quality filtering procedures, there is a small difference in the number of subjects in each subgroup. For both Method 1 and Method 2, the classification accuracy on our datasets is marginally better than those on the datasets provided by [34] in all subgroups except for "Children". The difference in performance is likely due to the slight difference in the number of subjects in the two datasets. The results validate that the quality of the brain networks generated by our pipeline is up to the standard of those used in other studies.

## 5.4 Sensitivity Study on Number of ROIs and Training Set Size

To assess the effect of the number of ROIs to model performance, we tune the number of ROIs on the Schaefer parcellation from {100, 200, 500, 1000} on the classification task. The results are reported in Table 7. Each model performs the best when the number of ROIs is set at 100 or 200.

We also test on different training set sizes by tuning $k$ in $k$-fold CV and report the results in Figure 3. The performance of each method improves with the increase in the training set size (larger $k$), which confirms the necessity of having datasets at this scale.

## 6 Conclusions, Limitations, and Future Directions

We release a functional brain network collection to the public at a large scale. The collection originates from 6 raw MRI image sources, covers 4 brain conditions, and totals to 2,702 subjects. Working with domain experts, we come up with a unified pipeline that converts raw fMRI and T1w images to brain networks. We tested the collection on 12 ML models and a recent graph analysis model to demonstrate that the data quality is not compromised while at the same time providing the results as domain baselines. We hope that the release of this collection of brain networks, together with the complete code of the processing pipeline, will promote both the development of graph-based models and the clinical advancement in the diagnosis and early intervention of neurodegenerative diseases.

We identify two limitations of this study: (1) Though the current brain network collection contains 2,702 subjects, it may still be considered small when applied to certain machine learning models (*e.g.*, deep learning models). Due to the high computational cost, we are not able to produce a larger collection for now but will make continuous efforts in enriching the collection, (2) The current collection focuses on extracting brain networks from a single modality, that is, the functional MRI images. Recently, neuroimaging data lean towards the direction of combining different modalities such as diffusion and functional MRI for further integrated explorations of brain functionalities. The fusion of functional MRI and diffusion MRI can encode information regarding integrated activities of the human brain. In the future, we plan to work on the processing pipeline of diffusion MRI images to make multi-modal brain networks publicly accessible to the research community.

## Acknowledgments

This research is supported by the MBIE Catalyst Strategic Fund UOAX2001 under the NZ-Singapore Data Science Research Programme, New Zealand, and the National Research Foundation, Singapore under its Industry Alignment Fund – Pre-positioning (IAF-PP) Funding Initiative. Any opinions, findings and conclusions or recommendations expressed in this material are those of the author(s) and do not reflect the views of National Research Foundation, Singapore.

Neurocon data used in this article was obtained from the NEUROCON project (84/2012), financed by UEFISCDI.

ADNI data collection used in this article was funded by the ADNI (National Institutes of Health Grant U01 AG024904) and DOD ADNI (Department of Defense award number W81XWH-12-2-0012). ADNI is funded by the National Institute on Aging, the National Institute of Biomedical Imaging and Bioengineering, and through generous contributions from the following: AbbVie, Alzheimer's Association; Alzheimer's Drug Discovery Foundation; Araclon Biotech; BioClinica, Inc.; Biogen; Bristol-Myers Squibb Company; CereSpir, Inc.; Cogstate; Eisai Inc.; Elan Pharmaceuticals, Inc.; Eli Lilly and Company; EuroImmun; F. Hoffmann-La Roche Ltd and its affiliated company Genentech, Inc.; Fujirebio; GE Healthcare; IXICO Ltd.; Janssen Alzheimer Immunotherapy Research & Development, LLC.; Johnson & Johnson Pharmaceutical Research & Development LLC.; Lumosity; Lundbeck; Merck & Co., Inc.; Meso Scale Diagnostics, LLC.; NeuroRx Research; Neurotrack Technologies; Novartis Pharmaceuticals Corporation; Pfizer Inc.; Piramal Imaging; Servier; Takeda Pharmaceutical Company; and Transition Therapeutics. The Canadian Institutes of Health Research is providing funds to support ADNI clinical sites in Canada. Private sector contributions are facilitated by the Foundation for the National Institutes of Health (www.fnih.org). The grantee organization is the Northern California Institute for Research and Education, and the study is coordinated by the Alzheimer's Therapeutic Research Institute at the University of Southern California. ADNI data are disseminated by the Laboratory for Neuro Imaging at the University of Southern California.

Mātai data collection was supported by Kānoa, New Zealand; the Hugh Green Foundation; an anonymous donation; and the MBIE Catalyst Strategic Fund UOAX2001 under the NZ-Singapore Data Science Research Programme, New Zealand. We are grateful to the Tairāwhiti Gisborne community and Mātai Ngā Māngai Māori for their guidance. We would like to sincerely thank our research participants for dedicating their time toward and support for this study. We would like to acknowledge the support from the Mātai Medical Research Institute imaging team, including Paul Condron, Taylor Emsden, Dr Patrick McHugh, Leigh Potter, Dr Samantha Holdsworth, Davidson Taylor, Dr Maryam Tayebi, Dr Daniel Cornfeld, and to the Mātai interns for assisting with the data collection and related community outreach and education activities. We are also grateful for the support of Dr Jerome Maller from GE Healthcare for his guidance on the imaging protocol.

PPMI data collection used in this article were obtained from the PPMI database (https://www.ppmi-info.org/accessdata-specimens/download-data). PPMI – a public-private partnership – is funded by The Michael J. Fox Foundation for Parkinson's Research and funding partners, including 4D Pharma, AbbVie Inc., AcureX Therapeutics, Allergan, Amathus Therapeutics, Aligning Science Across Parkinson's (ASAP), Avid Radiopharmaceuticals, Bial Biotech, Biogen, BioLegend, Bristol Myers Squibb, Calico Life Sciences LLC, Celgene Corporation, DaCapo Brainscience, Denali Therapeutics, The Edmond J. Safra Foundation, Eli Lilly and Company, GE Healthcare, GlaxoSmithKline, Golub Capital, Handl Therapeutics, Insitro, Janssen Pharmaceuticals, Lundbeck, Merck & Co., Inc, Meso Scale Diagnostics, LLC, Neurocrine Biosciences, Pfizer Inc., Piramal Imaging, Prevail Therapeutics, F. Hoffmann-La Roche Ltd and its affiliated company Genentech Inc., Sanofi Genzyme, Servier, Takeda Pharmaceutical Company, Teva Neuroscience, Inc., UCB, Vanqua Bio, Verily Life Sciences, Voyager Therapeutics, Inc., Yumanity Therapeutics, Inc.. For up-to-date information on the study, visit www.ppmi-info.org.

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

# A Appendix

## A.1 Dataset-related Details

### A.1.1 Dataset Documentation

We provide a datasheet that documents this data collection (in the Supplementary materials).

### A.1.2 Access Link and Data Repositories

The brain network datasets can be accessed at: `https://doi.org/10.17608/k6.auckland.21397377`.

We have a Github repository at `https://github.com/brainnetuoa/data_driven_network_neuroscience` which includes all our preprocessing codes and a demo on how to convert a raw fMRI image to a brain network in a step-by-step manner with sample input images from a subject in TaoWu.

**Figshare** (`https://www.figshare.com`) is a citable, shareable, and discoverable open data repository used by many educational institutions around the world, including ours. Figshare fits the many requirements of NeurIPS, including long term preservations of the data, access to a persistent identifier (an assigned DOI), and clear licensing and data standards.

### A.1.3 Author Statement

As authors, we confirm that we bear all responsibility in case of any violation of rights during the collection of the data or other work, and will take appropriate action when needed, to remove data with such issues.

### A.1.4 License of our Brain Network Datasets

CC BY-NC-SA 4.0

### A.1.5 Consent Process for Neuroimages

ABIDE, TaoWu, and Neurocon all follow Creative Common licenses and their source neuroimages are open use. For ADNI and PPMI we have followed their policies (Data Usage and Publication Policy) on how to use and publish derived data and where appropriate, we have submitted our manuscripts for permission to publish. Both required proper acknowledgements of the entity and their funders in the paper, which we have included.

### A.1.6 Computing Resources Used for Preprocessing

As stated in the main paper, fMRIPrep is computationally expensive. In our case, one run on one subject of fMRIPrep (with surface reconstruction enabled) using 8 threads on an Intel(R) Core(TM) i9-10940X CPU @ 3.30GHz takes on average 4-5 hours. Preprocessing of all data sources was split between two sites running on different internal cluster configurations. We are currently storing approximately 5 TB of data on these clusters, including the original neuroimages, preprocessed neuroimages, and our derived brain network data.

Site 1 cluster:

1. 3x Intel(R) Core(TM) i9-10940X CPU @ 3.30GHz

2. 1x Intel(R) Xeon(R) CPU ES-4020 v2 @ 2.60GHz

Site 2 cluster:

1. 5x Intel(R) Xeon(R) Silver 4314 CPU @ 2.40GHz

2. 1x Intel(R) Xeon(R) Gold 6230 CPU @ 2.10GHz

## A.2 Experiment-related Details

### A.2.1 Code

The code we used to run our entire pipeline and baseline experiments can be accessed on our Github: `https://github.com/brainnetuoa/data_driven_network_neuroscience`.

### A.2.2 Parameters

As stated in the main paper we used scikit-learn for all our implementations and here we provide the full list of parameters (including those we used for Grid Search).

**LogisticRegression**(*penalty:('l1', 'l2', 'elasticnet', 'none'), dual=False, tol=0.0001, C=1.0, fit_intercept=True, intercept_scaling=1, class_weight=None, random_state=None, solver='lbfgs', max_iter=1000000, multi_class='auto', verbose=0, warm_start=False, n_jobs=None, l1_ratio=None)*

**GaussianNB**(*priors=None, var_smoothing=1e-09)*

**SVC**(*C=1, kernel=('rbf','linear','poly','sigmoid','shortest path'), degree=3, gamma='auto', coef0=0.0, shrinking=True, probability=False, tol=0.001, cache_size=200, class_weight=None, verbose=False, max_iter=- 1, decision_function_shape='ovr', break_ties=False, random_state=None)*

**KNeighborsClassifier**(*n_neighbors=5, weights=('uniform', 'distance'), algorithm='auto', leaf_size=30, p=(1,2), metric='minkowski', metric_params=None, n_jobs=None)*

**RandomForestClassifier**(*n_estimators=(50,100,150,200), criterion= 'entropy', max_depth=5, min_samples_split=2, min_samples_leaf=1, min_weight_fraction_leaf=0.0, max_features='sqrt', max_leaf_nodes=None, min_impurity_decrease=0.0, bootstrap=True, oob_score=False, n_jobs=None, random_state=0, verbose=0, warm_start=False, class_weight=None, ccp_alpha=0.0, max_samples=None)*

### A.2.3 Computing Resources Used for Baselines

Experiments on baselines were performed on the internal cluster at site 2 (stated above).

### A.2.4 Connectivity Measure Comparison on TaoWu

Table 8: Connectivity Measure Comparison - Classification Accuracy (mean±standard deviation)

| | TaoWu - AAL116 | | | |
| --- | --- | --- | --- | --- |
| | Correlation | Covariance | Partial Correlation | Precision |
| Logistic Regression | 82.5±12.7 | 82.5±12.7 | 62.5±5.0 | 65.0±12.2 |
| Naïve Bayes | 61.5±19.0 | 61.5±19.0 | 45.0±16.2 | 43.5±16.2 |
| SVC | 85.0±9.4 | 85.0±9.4 | 65.0±20.0 | 65.0±14.6 |
| *k*NN | 60.0±21.5 | 60.0±14.6 | 52.5±5.0 | 57.5±6.1 |
| Random Forest | 72.5±12.2 | 70.0±10.0 | 65.0±18.4 | 62.5±7.9 |

