# Datasheet for Data-Driven Network Neuroscience: On Data Collection and Benchmark

## I. Motivation

### A. For what purpose was the dataset created? Was there a specific task in mind? Was there a specific gap that needed to be filled?

Graph is an ideal model for human neural data. With the expansion in the size, scope and complexity of human neural data in recent years, making a large collection of *brain network datasets* available to the public becomes important to unleash the potential of network neuroscience. Our released data collection aims to fill this gap and lower the barrier to entering this interdisciplinary field, with the hope to promote the research in graph-based analytical and clinical studies such as the detection of neurodegenerative conditions.

### B. Who created the dataset (e.g., which team, research group) and on behalf of which entity (e.g., company, institution, organization)?

The authors of the paper collected the MRI images from 5 public databases and one private data source (Mātai Medical Research Institute). They worked with domain experts to make sensible design choices, and pre-processed the MRI images to produce a collection of brain network datasets.

Specifically, the gathering and processing of MRI images from the 5 public databases were conducted by David Tse Jung Huang and Sophi Shilpa Gururajapathy, with the help of Yiping Ke, Miao Qiao, Alan Wang, Haribalan Kumar, Yunhan Yang, and Jiaxing Xu. Involved institutions and organizations include the University of Auckland, Mātai, and Nanyang Technological University.

The gathering of the MRI images of the Mātai data was performed by Eryn Kwon and Josh McGeown with the help of Paul Condron, with the data processing performed by Yunhan Yang. Involved institutions/organizations include Mātai and the University of Auckland.

### C. Who funded the creation of the dataset? If there is an associated grant, please provide the name of the grantor and the grant name and number.

This work is funded by MBIE Catalyst - Strategic Fund NZ-Singapore Data Science Research Programme UOAX2001 and National Research Foundation, Singapore under its Industry Alignment Fund – Pre-positioning (IAF-PP) Funding Initiative.

Funding sources for collecting the Mātai dataset include the Catalyst Strategic Fund from Government Funding administered by the New Zealand Ministry of Business Innovation and Employment, Kānoa Regional Economic Development & Investment Unit, New Zealand, and the Hugh Green Foundation.

## II. Composition

### A. What do the instances that comprise the dataset represent (e.g., documents, photos, people, countries)? Are there multiple types of instances (e.g., movies, users, and ratings; people and interactions between them; nodes and edges)?

This collection consists of the brain networks of a number of subjects, where each subject represents a person (could be healthy or with some brain condition). Each brain network has regions-of-interest (ROIs) as nodes and the correlation between BOLD signals of ROIs as edges. The BOLD signals of ROIs are also presented as node features.

### B. How many instances are there in total (of each type, if appropriate)?

This collection has 6 datasets with a total of 2,702 subjects. Each dataset has a different number of subjects as in Table I. Each subject has been parcellated with 5 Parcellation Methods (PMs): AAL, HarvardOxford (HO), Schaefer, k-means, and Ward, resulting in a total of 13,510 brain networks in this collection.

Each subject in each dataset is labeled with a specific class. The class distribution is provided in Table II.

TABLE I
OUR COLLECTION OF BRAIN NETWORK DATASETS: SUMMARY

| Dataset | Condition | # Subjects | # Classes |
|---|---|---|---|
| ABIDE | Autism Spectrum Disorder | 1,025 | 2 |
| ADNI | Alzheimer's Disease | 1,327 | 6 |
| PPMI | Parkinson's Disease | 209 | 4 |
| Mātai | mTBI | 60 | 2 |
| TaoWu | Parkinson's Disease | 40 | 2 |
| Neurocon | Parkinson's Disease | 41 | 2 |

### C. Does the dataset contain all possible instances or is it a sample (not necessarily random) of instances from a larger set? If the dataset is a sample, then what is the larger set? Is the sample representative of the larger set (e.g., geographic coverage)? If so, please describe how this representativeness was validated/verified. If it is not representative of the larger set, please describe why not (e.g., to cover a more diverse range of instances, because instances were withheld or unavailable).

Not all images contained in all data sources are included in our collection. We filter the subjects based on the availability,

TABLE II

OUR COLLECTION OF BRAIN NETWORK DATASETS: CLASS
DISTRIBUTION

| Dataset | Class | # Subjects |
|---------|-------|-----------|
| ABIDE | Control | 537 |
| | ASD | 488 |
| ADNI | CN | 819 |
| | SMC | 73 |
| | LMCI | 102 |
| | EMCI | 89 |
| | MCI | 179 |
| | AD | 65 |
| PPMI | Control | 15 |
| | SWEDD | 14 |
| | Prodromal | 67 |
| | PD | 113 |
| Mātai | Pre-season | 35 |
| | Post-season | 25 |
| TaoWu | Control | 20 |
| | PD | 20 |
| Neurocon | Control | 15 |
| | PD | 26 |

TABLE III

DATASET STATISTICS IN SOURCES AND OUR COLLECTION. GENDER
DISTRIBUTION IS SHOWN IN PARENTHESES (MALE/FEMALE).

| Dataset | # Subjects in Source | # Subjects in Our Collection |
|---------|---------------------|------------------------------|
| ABIDE | 1,110 (946/164) | 1,025 (873/152) |
| ADNI | 1,399 (640/759) | 1,327 (599/728) |
| PPMI | 213 (130/83) | 209 (127/82) |
| Mātai | 60 (60/0) | 60 (60/0) |
| TaoWu | 40 (23/17) | 40 (23/17) |
| Neurocon | 43 (21/22) | 41 (19/22) |

validity, and the quality of our required MRI image types (rs-fMRI and T1w).

For ABIDE, TaoWu, and Neurocon, we include all subjects from the original sources except for those that had quality issues. A subject has quality issues if it has an incomplete image (i.e., not containing the full brain) and/or damaged data (i.e., with an error reported by any subsequent preprocessing steps).

For longitudinal study data sources of ADNI and PPMI, each subject typically has multiple scans. We use the baseline study (the first scan) of each subject: as suggested by our domain experts, the baseline study is usually the most comprehensive one that would cover a wide range of modalities.

For the Mātai dataset, the post-season MRI scan collection was performed on the subjects who had pre-season scans. The reduction in the subject number is due to attrition, or a change in their eligibility status.

Available metadata of each subject from the source is included in our collection, such as age and gender. The gender distributions of our datasets closely match those of the original data sources. The only exception is ADNI: our released data contains 0.6% more females than the original ADNI data. The statistics comparison between the sources and our collection is provided in Table III[1].

### D. What data does each instance consist of?

One subject under one PM has two .mat files.

- A weighted connectivity matrix, with each entry representing the correlation values computed on the BOLD signals between two ROIs. For example, the AAL atlas was one of the PMs used and it parcellates the brain into 116 ROIs. Therefore, the .mat connectivity matrix file derived with AAL has a dimension of $116 \times 116$.

[1] Raw data of ADNI and PPMI were downloaded in April 2022. The # Subjects in Source in Table III are with respect to this timestamp. Note that for PPMI we preprocessed each subject's first scan that captured both rs-fMRI and T1w images on the same day.

- A node feature matrix, with each row storing the aggregated BOLD signal for each ROI. The aggregation was computed as the average BOLD signal of all the voxels contained in the ROI. The dimension of the BOLD signal is determined by the length of the rs-fMRI scan and varies in different datasets.

### E. Is there a label or target associated with each instance?

There is a class label associated with each subject. The class distribution of each dataset is shown in Table II. Subjects in our collection are organized in different folders with the folder name in the form of sub-{SubjectClass}{ID}. SubjectClass indicates the class label of the subject, and ID refers to the unique ID number assigned to each subject by the neuroimaging study.

### F. Is any information missing from individual instances? If so, please provide a description, explaining why this information is missing (e.g., because it was unavailable). This does not include intentionally removed information, but might include, e.g., redacted text.

No information is missing.

### G. Are relationships between individual instances made explicit (e.g., users' movie ratings, social network links)? If so, please describe how these relationships are made explicit.

None.

### H. Are there recommended data splits (e.g., training, development/validation, testing)? If so, please provide a description of these splits, explaining the rationale behind them.

None. In our experimental study on this collection, we split each dataset into 8:1:1 for training, validation and test with 10-fold cross-validation on each dataset, but this is not necessarily the standard. Other data splits on the collection are possible.

### I. Are there any errors, sources of noise, or redundancies in the dataset? If so, please provide a description.

As with all MRI images, there is inevitably noise contained in the scans. We have followed domain standards for removing noises in the original scans. We adopted the standard 9 parameter (9P) confound setting widely used in functional connectivity studies: white matter, cerebrospinal fluid, global signal, and the 6 rigid-body motion parameters for rotation and translations at $x$, $y$ and $z$ axes.

*J. Is the dataset self-contained, or does it link to or otherwise rely on external resources (e.g., websites, tweets, other datasets)? If it links to or relies on external resources, a) are there guarantees that they will exist, and remain constant, over time; b) are there official archival versions of the complete dataset (i.e., including the external resources as they existed at the time the dataset was created); c) are there any restrictions (e.g., licenses, fees) associated with any of the external resources that might apply to a dataset consumer? Please provide descriptions of all external resources and any restrictions associated with them, as well as links or other access points, as appropriate.*

Our brain network data collection is self-contained for the networks originated from 6 public data sources of raw neuroimages: we have included in this collection the complete set of brain networks, as well as their corresponding preprocessed neuroimages. As requested by the owner of ADNI, the derived ADNI dataset needs to be hosted on LONI IDA whose access point will be listed on the Figshare repository upon the data release. For the Mātai dataset, no direct identification of the individuals from the released data is possible.

## III. COLLECTION PROCESS

*A. How was the data associated with each instance acquired? Was the data directly observable (e.g., raw text, movie ratings), reported by subjects (e.g., survey responses), or indirectly inferred/derived from other data (e.g., part-of-speech tags, model-based guesses for age or language)? If the data was reported by subjects or indirectly inferred/derived from other data, was the data validated/verified? If so, please describe how.*

The data we produced for each instance is the brain network converted from raw MRI images. The sources of the raw MRI images are discussed in Section 3 and the conversion process is described in Section 4 of the main paper.

*B. What mechanisms or procedures were used to collect the data (e.g., hardware apparatus or sensor, manual human curation, software program, software API)? How were these mechanisms or procedures validated?*

Raw neuroimages from 5 public data sources were acquired online from their respective repositories. For detailed scanning and acquisition protocols of these neuroimages, please refer to the descriptions in each data source.

The Mātai dataset was collected using a 3.0 T MR scanner (GE SIGNA Premier; General Electric, MI, USA) with a 48-channel head coil. The sequences processed from the Mātai dataset were 3D T1-weighted, sagittal 3D T2-FLAIR (Fluid-Attenuated Inversion Recovery), and resting-state functional MRI. The MRI data is stored in a DICOM format in a secure file repository with restricted access.

*C. Who was involved in the data collection process (e.g., students, crowdworkers, contractors) and how were they compensated (e.g., how much were crowdworkers paid)?*

The collection process of the raw neuroimages from public data sources varies for each study and is detailed by each study.

The Mātai dataset was created through a collaboration between the University of Auckland and Mātai Medical Research Institute. Research Fellows were compensated for the study design, implementation, fieldwork, data collection, data curation, and image processing. The cost of the MRI sequences used in the paper was covered by our MBIE grant.

*D. Over what timeframe was the data collected? Does this timeframe match the creation timeframe of the data associated with the instances (e.g., recent crawl of old news articles)? If not, please describe the timeframe in which the data associated with the instances was created.*

For ABIDE, TaoWu, and Neurocon, all the raw neuroimages provided by the data source were used. For longitudinal studies of ADNI and PPMI, the raw neuroimages were downloaded in April 2022.

The Mātai data was created over a single season of rugby between the timeframe of April-September of one year (the same year).

## IV. DATA PREPROCESSING

*A. Was any preprocessing/cleaning/labeling of the data done (e.g., discretization or bucketing, tokenization, part-of-speech tagging, SIFT feature extraction, removal of instances, processing of missing values)? If so, please provide a description. If not, you may skip the remainder of the questions in this section.*

This data collection went through a full preprocessing pipeline. Details are provided in Section 4 of the main paper.

*B. Was the "raw" data saved in addition to the preprocessed/cleaned/labeled data (e.g., to support unanticipated future uses)? If so, please provide a link or other access point to the "raw" data.*

For the 5 public data sources, we are not able to include the raw neuroimages in our released collection: the raw data need to be accessed via the repositories of the sources, according to their data policy. Download links are provided below.

- ABIDE: `https://ida.loni.usc.edu/login.jsp`
- ADNI: `https://adni.loni.usc.edu/data-samples/access-data/`
- PPMI: `https://www.ppmi-info.org/access-data-specimens/download-data`
- TaoWu: `https://fcp-indi.s3.amazonaws.com/data/Projects/INDI/umf_pd/taowu.tar.gz`
- Neurocon: `https://fcp-indi.s3.amazonaws.com/data/Projects/INDI/umf_pd/neurocon.tar.gz`

We are not releasing the raw data of the Matāi to respect Māori data sovereignty, which emphasizes the right of indigenous communities to control and manage the use and sharing of their own data according to their specific cultural and ethical considerations.

*C. Is the software used to preprocess/clean/label the instances available? If so, please provide a link or other access point.*

Yes. Our preprocessing codes and a demo are available at `https://github.com/brainnetuoa/data_driven_network_neuroscience`.

*D. Does this dataset collection/processing procedure achieve the motivation for creating the dataset stated in the first section of this datasheet? If not, what are the limitations?*

Yes.

## V. DATASET DISTRIBUTION

*A. Will the dataset be distributed to third parties outside of the entity (e.g., company, institution, organization) on behalf of which the dataset was created? If so, please provide a description.*

Yes, our brain network collection will be made publicly accessible.

*B. How will the dataset be distributed (e.g., tarball on website, API, GitHub)? Does the dataset have a digital object identifier (DOI)?*

The data collection with Creative Common licenses will be distributed via Figshare with a DOI at `https://doi.org/10.17608/k6.auckland.21397377`. Per request of the raw neuroimage owners of ADNI, the derived ADNI dataset will be hosted on LONI IDA.

The preprocessing codes and a demo are available at `https://github.com/brainnetuoa/data_driven_network_neuroscience`.

*C. When will the dataset be released/first distributed? What license (if any) is it distributed under? Are there any copyrights on the data? Will the dataset be distributed under a copyright or other intellectual property (IP) license, and/or under applicable terms of use (ToU)?*

The brain network collection will be released upon acceptance of the paper and apart from ADNI, all datasets will be released under CC BY-NC-SA 4.0 license. ADNI will remain under ADNI's licensing policy and will comply with LONI IDA's data sharing policy.

*D. Do any export controls or other regulatory restrictions apply to the dataset or to individual instances? If so, please describe these restrictions, and provide a link or other access point to, or otherwise reproduce, any supporting documentation.*

None for the brain network collection.

*E. Are there any fees or access/export restrictions?*

No, we do not impose any fees for accessing our brain network collection.

## VI. DATASET MAINTENANCE

*A. Who will be supporting/hosting/maintaining the dataset?*

Yiping Ke and Miao Qiao will be supporting and maintaining the datasets. The datasets are hosted in Figshare and Figshare will be handling the long-term accessibility of the data.

*B. How can the owner/curator/manager of the dataset be contacted (e.g., email address)?*

The curators of this collection, Yiping Ke and Miao Qiao, can be contacted at `ypke@ntu.edu.sg` and `miao.qiao@auckland.ac.nz`, respectively.

*C. Is there an erratum? If so, please provide a link or other access point.*

Not at this stage. We will maintain an erratum upon the public release of the data collection.

*D. Will the dataset be updated (e.g., to correct labeling errors, add new instances, delete instances)? If so, please describe how often, by whom, and how updates will be communicated to dataset consumers (e.g., mailing list, GitHub)?*

For the functional brain networks in the current collection, we will update the datasets whose data sources continuously make new subjects available (e.g., PPMI). We also plan to enrich the collection with brain networks from additional MRI modularities (Diffusion is planned) and parcellations in the future. It will be updated by the same team that created this data collection. Diffusion-based brain networks are planned to be released by Dec 2024.

The update will be announced on our project site `https://doi.org/10.17608/k6.auckland.21397377`, on which users can choose to stay informed about the new updates of our collection by providing their emails to our mailing list.

*E. Is there a repository to link to any/all papers/systems that use this dataset?*

Not at this stage.

*F. If the dataset relates to people, are there applicable limits on the retention of the data associated with the instances (e.g., were the individuals in question told that their data would be retained for a fixed period of time and then deleted)? If so, please describe these limits and explain how they will be enforced.*

The collection of brain networks derived from public databases does not have a retention limitation.

Matāi has a minimum retention limitation of 10 years on the raw data and no maximum retention limitation.

*G. Will older versions of the dataset continue to be supported/hosted/maintained? If so, please describe how. If not, please describe how its obsolescence will be communicated to dataset consumers.*

Yes. We are releasing our data on Figshare which has the function of version control. Once published, the item DOI will version with every change, update or edit to the file or metadata record. An updated dataset is an appropriate reason for versioning and all previous versions will remain publicly accessible.

*H. If others want to extend/augment/build on/contribute to the dataset, is there a mechanism for them to do so? If so, please provide a description. Will these contributions be validated/verified? If so, please describe how. If not, why not? Is there a process for communicating/distributing these contributions to dataset consumers? If so, please provide a description.*

Not at this stage, but researchers are encouraged to contact the authors for such extensions.

## VII. Legal and Ethical Considerations

*A. Were any ethical review processes conducted (e.g., by an institutional review board)? If so, please provide a description of these review processes, including the outcomes, as well as a link or other access point to any supporting documentation.*

For public sources, we have obtained consents from data sources in using their data and publishing the paper.

For the Mātai dataset, data were acquired under informed consent and ethical approval (HDEC ethics approval 20/NTB/14). It involved significant consultation around the distribution of the Mātai brain network data, via informed consent and on-going engagement and consultation with the research participants. Due care was taken by notifying the stakeholders (Tairāwhiti Gisborne community groups, Ngā Māngai Māori, and research participants) regarding our intention to publish any outcomes related to the dataset (both academic and non-academic).

*B. Does the dataset contain data that might be considered confidential (e.g., data that is protected by legal privilege or by doctor–patient confidentiality, data that includes the content of individuals' non-public communications)? If so, please provide a description.*

No.

*C. Does the dataset contain data that, if viewed directly, might be offensive, insulting, threatening, or might otherwise cause anxiety? If so, please describe why.*

No.

*D. Does the dataset relate to people? If not, you may skip the remaining questions in this section.*

Yes.

*E. Does the dataset identify any subpopulations (e.g., by age, gender)?*

Some datasets in the collection have meta-data of subjects available and we have included them. Based on the meta-data (e.g., age, gender, eyes-closed), subpopulations could be identified. Please see an example of using such subpopulations in our experiments on ABIDE (Section 5.2 in the main paper). The respective distributions are provided in Table 4 of the main paper.

*F. Is it possible to identify individuals (i.e., one or more natural persons), either directly or indirectly (i.e., in combination with other data) from the dataset? If so, please describe how.*

No. Sensitive subject information has already been removed from original data sources, i.e., all data are de-identified. In particular, no direct identification of the individuals is possible for the Mātai dataset.

*G. Does the dataset contain data that might be considered sensitive in any way (e.g., data that reveals racial or ethnic origins, sexual orientations, religious beliefs, political opinions or union memberships, or locations; financial or health data; biometric or genetic data; forms of government identification, such as social security numbers; criminal history)? If so, please provide a description.*

No. For the Matāi dataset, we didn't include any metadata or raw imaging data in the release.

*H. Did you collect the data from the individuals in question directly, or obtain it via third parties or other sources (e.g., websites)?*

Only the Mātai dataset was collected from individuals; the raw data of other datasets were obtained from the data repositories of their original neuroimaging studies.

*I. Were the individuals in question notified about the data collection? If so, please describe (or show with screenshots or other information) how notice was provided, and provide a link or other access point to, or otherwise reproduce, the exact language of the notification itself.*

Yes. The participant information sheet for the Mātai dataset is provided as part of the consenting process and is available at `https://redcap.fmhs.auckland.ac.nz/surveys/?s=DWA8ACYNN3`.

*J. Did the individuals in question consent to the collection and use of their data? If so, please describe (or show with screenshots or other information) how consent was requested and provided, and provide a link or other access point to, or otherwise reproduce, the exact language to which the individuals consented.*

Yes. Mātai data was collected with informed consent/assent approved by (HDEC ethics approval 20/NTB/14). The electronic consent form is available at: `https://redcap.fmhs.auckland.ac.nz/surveys/?s=DWA8ACYNN3`.

*K. If consent was obtained, were the consenting individuals provided with a mechanism to revoke their consent in the future or for certain uses? If so, please provide a description, as well as a link or other access point to the mechanism (if appropriate).*

Yes, as part of the consenting process, the individuals were notified of the mechanism and time frame to revoke their consent (`https://redcap.fmhs.auckland.ac.nz/surveys/?s=DWA8ACYNN3`).

*L. Has an analysis of the potential impact of the dataset and its use on data subjects (e.g., a data protection impact analysis) been conducted? If so, please provide a description of this analysis, including the outcomes, as well as a link or other access point to any supporting documentation.*

No.

## VIII. Uses

*A. Has the dataset been used for any tasks already? If so, please provide a description.*

The collection of brain networks has been used in this paper for the task of brain condition prediction.

*B. What (other) tasks could the dataset be used for?*

The data collection was created for brain network analysis and clinical tasks.

*C. Is there anything about the composition of the dataset or the way it was collected and preprocessed/cleaned/labeled that might impact future uses? For example, is there anything that a dataset consumer might need to know to avoid uses that could result in unfair treatment of individuals or groups (e.g., stereotyping, quality of service issues) or other risks or harms (e.g., legal risks, financial harms)? If so, please provide a description. Is there anything a dataset consumer could do to mitigate these risks or harms?*

We are not aware of risks such as unfair treatment of individuals or groups (e.g., stereotyping, quality of service issues) or other risks or harms (e.g., legal risks, financial harms): for Matāi data, we didn't include any metadata nor raw imaging data in the release; for other brain network datasets, we derived them from public databases; all the brain networks in the collection were produced via a unified preprocessing pipeline.

*D. Are there tasks for which the dataset should not be used? If so, please provide a description.*

The collection consists of 6 datasets of functional brain networks. They represent human brains as graphs and should not be used for non-graph-based studies or studies that do not intend to use functional brain data.