# OpenReview forum: "Data-Driven Network Neuroscience: On Data Collection and Benchmark"
_NeurIPS.cc/2023/Track/Datasets_and_Benchmarks — NeurIPS 2023 Datasets and Benchmarks Poster_

### Official Review · Reviewer_v8Nt · 2023-07-14
**Valuable dataset saving neuroimaging researchers a lot of work**

**Rating:** 7
**Confidence:** 3
**Correctness:** As far as I can judge, everything is …

**Strengths:**

The methodology in this paper is sound and well described. The paper is overall well written and contains sufficient details. Data and code are provided in publicly accessible, versioned and findable repositories and well documented. A complete data sheets is provided containing a lot of a lot of additional information. Overall, a lot of thought and effort has been put into this dataset and I am certain that it will prove useful for the community.

**Additional Feedback:**

None right now.

**Clarity:**

Yes, it is. A few things are unclear though:

1.	“We acknowledge that all data has an origin of significance (whakapapa).”
2.	The paragraph on page 5 starting with “For validity”. How do you make sure that an image conforms to a label such as “ADHD” or “normal”? It should not be obvious. What do you mean by “the image databases often had inconsistent labeling”


**Documentation:**

The dataset is described in detail and moreover an extensive data sheet is provided in the supplement. Data and code are provided in open repositories. All relevant issues regarding ethical use, data protection, licensing, etc. are discussed as far as I can tell.

**Ethics:**

No specific ethical concerns have been identified.

**Limitations:**

I do not see major limitations of this work. It may be worth pointing out so, that almost anybody could have compiled that dataset. Apart from the MATAI data, all data used here are already in the public domain. The authors process these data with openly available pipelines. So, overall, there is no strong intellectual contribution from the authors here. Nevertheless, it should be a very valuable dataset, as these are complex pipelines that need to be applied in the right way. And the availability of the processed (connectomic) data could save a lot of researchers tedious work and computing resources.

**Opportunities For Improvement:**

I do not see much necessity for improvement. One thing, though, is that the authors repeatedly refer to a group of domain experts who have guided the various steps and design choices. It would be good to make this a little bit more concrete. How was this group composed, what was the process through which the feedback was provided. What are examples in which the feedback by the domain experts led to design changes. It is mentioned that MRI experts performed quality checks in the fMRIprep outputs. How was this organized. Was every single output checked this way? Can you elucidate what the criteria imposed by the experts were?

It is also mentioned that the exemplary classification analysis was made to “assess if the graph representation maintains the data quality from neuroimages”. Thus, I would have expected that the classification based on FC features would be compared to a classification based on BOLD time series.

**Relation To Prior Work:**

This work makes use of existing public dataset as well as processing pipelines, which are all properly cited.

**Summary And Contributions:**

The authors present a large dataset of functional brain connectomes (FC) derived from functional MRI. The dataset consists of 6 subsets, which are (except one) all derived from different already existent large open fMRI studies. The raw data in these repositories has been preprocessed with state-of-the-art pipelines, quality checked, and further processed to yield brain region by region functional connectivity matrices/graphs (representing correlations across time). An exemplary machine learning analysis of the data is conducted using five standard methods, demonstrating above chance level prediction of different conditions of pathologies.

---

> ### Author Response · Authors · 2023-08-22
> **Response to Reviewer v8Nt**
>
> **[How domain experts guided the steps and design choices]** Further elaborations on domain expert engagement are outlined below. Regrettably, due to space constraints, the paper can only offer concise overviews, abstaining from extensive details.
>
> * Group of domain experts: composed of our co-authors Alan Wang and Haribalan Kumar. Alan is a professor at the Centre for Brain Research in the Faculty of Medical and Health Sciences and the Auckland Bioengineering Institute at the University of Auckland, New Zealand. Haribalan is an MRI scanning/preprocessing expert at General Electric Healthcare (GE), one of the top four brands of MRI scanners.
>
> * Feedback process/examples: The process entailed two main phases: (I) Designing the processing pipeline; and (II) Ensuring pipeline output quality. In phase I, we had intense discussions with our domain experts Alan and Hari centered on neuroscience processing norms, domain tools, and their merits. They steered fMRIPrep selection (see details in general response) and the adoption of commonly used parcellation schemes, covering representative schemes in both atlas-based and clustering-based categories. Phase II involved selecting scan samples from each source and submitting fMRIPrep outputs at each stage to Hari and Alan for meticulous quality checks. Manual inspection of output images and visualizations guided iterative evaluations—each step's approval hinged on Hari/Alan's endorsement. For instance, the surface reconstruction, not a default fMRIPrep setting, was enabled after their assessment to enhance data alignment across subjects, requiring an additional 3 computing hours per subject in preprocessing overhead.
>
> **[Comparison with classification based on BOLD time series]** Thank you for the suggestion. We have obtained the classification performance on two time series based models, GRU and 1D-CNN. The results are reported in the table below. Compared with both conventional and graph ML models, the performance based on BOLD signals is significantly worse. This verifies the effectiveness/usefulness of brain networks in the task of disease classification. The same conclusion was also drawn by FBNetGen [a]. We have added these results in Section 5.1 of the revision (Table 4).
>
> |               |     ABIDE           |     ADNI            |     PPMI             |     Matai            |     TaoWu            |     Neurocon         |
> |---------------|---------------------|---------------------|----------------------|----------------------|----------------------|----------------------|
> |     GRU       |     $53.6  \pm 1.4$    |     $61.8\pm 0.3$      |     $54.1\pm 2.2$       |     $58.3  \pm 8.3$     |     $50.0\pm 0.0$       |     $63.8  \pm 1.5$     |
> |     1D-CNN    |     $55.5  \pm 3.0$    |     $50.2  \pm 4.0$    |     $55.0  \pm 11.5$    |     $61.7  \pm 13.0$    |     $45.0  \pm 15.0$    |     $58.5  \pm 11.2$    |
>
> [a] Kan X, Cui H, Lukemire J, et al. Fbnetgen: Task-aware gnn-based fmri analysis via functional brain network generation[C]//International Conference on Medical Imaging with Deep Learning. PMLR, 2022: 618-637.
>
> **[Clarification on “We acknowledge that all data has an origin of significance (whakapapa).”]** This is to acknowledge that the data source and the participants in the Matai study are important. It is the requirement from the Maori advisory board to include this statement.
>
> **[Validity check on image labels]** Inconsistent subject label information was observed in downloaded files from data sources. Label details could exist in metadata files, folder names, or actual scan filenames, while label details may not align for the same subject. In cases where label disparities exist—like labeling a subject ASD in metadata but placing its scan in the control group folder—the subject's label is deemed invalid, leading to its removal from our data collection. To guarantee label consistency, we systematically validated subject labels across all data sources.

---

### Official Review · Reviewer_iqnx · 2023-07-16
**Looks okay**

**Rating:** 6
**Confidence:** 1
**Correctness:** I found no issues

**Strengths:**

The authors collect several existing datasets and make them easier to access. On the whole I appreciate the effort, but have many suggestions for improvement.

**Additional Feedback:**

N/A

**Clarity:**

The paper is readable, but the writing could be improved (see "Opportunities for Improvement")

**Documentation:**

As mentioned in "opportunities for improvement", I think the readme and documentation on figshare could be improved.

**Ethics:**

No concerns noted.

**Limitations:**

See "opportunities for improvement"

**Opportunities For Improvement:**

**Section 1.** The goal of this benchmark is not immediately clear from a quick read through the introduction. The first several paragraphs are very grandiose, and only briefly mentions the possibility to *"help identify subjects at different stages of neurological diseases"* in paragraph 3. **But as I understand it, this is the main point of the benchmark!** Why bury this message in a few short sentences in paragraph 3, before pivoting to a complex discussion of preprocessing?

If the authors intend for these datasets to be used for something other than classification, they should list more explicit examples in this first section and show examples later.

**Section 2.** This is mostly fine, but it is still unclear that the machine learning task of interest is to train classifiers to distinguish patients with disease from controls.

The clinical importance of this classification problem is not well-established in Section 2 or Section 1.

**Section 3.** Some details are missing. For instance, I assume that all of these datasets are "resting state" fMRI? Or is there a task each subject is performing in the scanner? These details should be specified and the limitations resting state fMRI discussed extensively.

**Section 4.** I'm not qualified to nitpick the design choices here. The authors seem to know what they are doing. Arguably, this benchmark would be better served at a venue other than NeurIPS where reviewers have more domain expertise.

**Section 5.** The "classification task" should be explained more explicitly. For example, the sentence *"The aim is to establish a basline study on this data collection"* -- which is not very informative to the reader. It would be better to say something like *"For each dataset, the aim is to classify affected patients from healthy control subjects"*.

Some of these benchmarks do not seem like a classification task. In particular, the Alzheimer's dataset has 6 categories with increasing severity of disease. [Ordinal regression](https://en.wikipedia.org/wiki/Ordinal_regression) would be a better way to frame that particular problem.

Finally, the accuracies in Table 3 are mostly underwhelming. It would be nice to explain somewhere (e.g. in Section 1 or 6) why solving this classification problem at this level of accuracy has the potential to be clinically useful.

**Released Benchmark.** It feels like a data dump on figshare. Having a website with more tutorials / documentation / walkthroughs would be useful.

**Relation To Prior Work:**

Yes, it is clearly discussed in Section 3. BUT - I lack enough domain expertise to be confident that the citations to prior work are comprehensive.

**Summary And Contributions:**

I have marked a low confidence score to reflect limited domain expertise.

The authors release pre-processed fMRI datasets, lowering the barrier to entry on some simple clinical classification tasks.

---

> ### Author Response · Authors · 2023-08-22
> **Response to Reviewer iqnx**
>
> **[Discussion on clinical importance of classification/ordinal regression]** Thank you for the suggestion. We have revised the first few paragraphs of Introduction (Page 2 of the revision) by shortening the background introduction and describing the clinical applications of classification/ordinal regression.
>
> **[Clarification of resting-state fMRI]** Thank you for bringing this to our attention. We have clarified in Section 3 and Table I that the datasets contain only resting-state data (Page 3 of the revision). Note that our preprocessing pipeline is not limited to resting-state data as fMRIPrep works generally for both resting-state and task-specific functional MRI images.
>
> **[Clarification on classification and inclusion of ordinal regression in experimental study]** We have revised the first paragraph of Section 5 (Page 7 of the revision) to state the aims of classification and ordinal regression. For the inclusion of ordinal regression results, please refer to the general response.
>
> **[Why solving classification at this level of accuracy has the potential to be clinically useful]** We have added the following discussions at the end of Section 5.1 (Page 9 of the revision). “The accuracy scores in Tables 3 and 4 might appear relatively modest, raising concerns about the clinical applicability of solving the classification problem. It's important to note that disease classification through brain networks is a nascent and evolving research domain. Many of the ML methods tested are general-purpose and not specifically designed for brain networks. Our release of fully weighted brain network matrices in this collection is anticipated to stimulate and advance research in this field. This initiative is expected to foster the development of tailored learning models for brain networks, yielding clinically relevant outcomes. The current results can serve as foundational benchmarks for future research endeavors.”
>
> **[Website with more tutorials/documentations/walkthroughs]** We have provided a Github repository at [https://github.com/bna-data-analysis/extract-brain-network](https://github.com/bna-data-analysis/extract-brain-network), which includes a demo on how to convert a raw fMRI image to a brain network in a step-by-step manner with sample input images from a subject in TaoWu.

---

### Official Review · Reviewer_ebHo · 2023-07-17
**fMRI data across several public datasets (ABIDE, ADNI, PPMI) and one private dataset (Matai).**

**Rating:** 6
**Confidence:** 4

**Strengths:**

●	Releasing a large-scale collection of brain networks helps address the lack of openly available data to drive research at the intersection of ML and neuroscience.
●	Multiple data sources covering different conditions like Autism, Alzheimer's, Parkinson's are included to enhance diversity.
●	Different parcellation schemes such as atlas-based and data-driven clustering are provided for the networks.
●	The conversion pipeline makes use of standard neuroscience software tools and incorporates domain expert input.
●	Classification experiments on 5 ML models validate the basic utility for machine learning tasks.


**Additional Feedback:**

The paper makes a useful contribution to releasing functional brain network data. However, the scope is limited to only one imaging modality currently. More extensive benchmarking and comparisons to prior datasets would highlight the value of this work.

**Clarity:**

The overall writing quality and structure are adequate but could be improved. The introduction and conclusion do not clearly summarize the limitations and potential future work. Certain aspects like parcellation schemes and their impact could also be explained more clearly.

**Correctness:**

The brain network construction methodology follows standard practices in the field. Both software tools and domain expert input are utilized. However, more extensive ablation studies evaluating the impact of key design choices would further validate correctness. The classification experiments appear technically sound but do not verify the data's integrity.



**Documentation:**

There is sufficient detail on data collection and organization, availability and maintenance, and ethical and responsible use.

**Ethics:**

No ethical concerns in this study.

**Limitations:**

●	The collection is restricted to functional connectivity networks from fMRI only.
●	No detailed analysis is provided on how the choice of parcellation scheme affects model performance.
●	The collection size is still relatively limited for effectively training and evaluating deeper graph neural network models.
●	Lack of comparisons to other existing neuroimaging datasets makes it difficult to gauge the value addition.




**Opportunities For Improvement:**

●	The collection is restricted to functional connectivity networks from fMRI only. Expanding to multimodal imaging data (e.g. diffusion MRI) could significantly increase scope and utility.
●	No detailed analysis is provided on how the choice of parcellation scheme affects model performance. This could provide useful insights.
●	Benchmarking on a more diverse set of graph analytics tasks beyond just classification would better characterize the data's potential.



**Relation To Prior Work:**

Relevant literature on neuroimaging collections, network construction pipelines, and graph-based analysis are suitably cited. The need for brain network data is well-established based on gaps in public datasets. However, more discussions situating this collection with respect to other available neuroimaging datasets could bring out the value addition.

**Summary And Contributions:**

This paper presents a collection of functional brain networks for 2,702 subjects, converted from raw fMRI data across several public datasets (ABIDE, ADNI, PPMI) and one private dataset (Matai). The conversion pipeline and some basic classification experiments are presented. The core contribution is releasing this functional connectivity data to enable graph-based neuroscience studies. However, the scope is currently limited to just one imaging modality and lacks extensive benchmarking.

---

> ### Author Response · Authors · 2023-08-22
> **Response to Reviewer ebHo**
>
> **[Expansion to multimodal imaging data]** Thank you for your suggestion. Our forthcoming objective is to enhance the existing collection by incorporating multimodal brain networks, and this endeavor is already in progress. We've successfully gathered DTI raw images, primarily from the same data sources and subjects as those featured in this study. Our current focus involves collaborating with domain experts to finalize the preprocessing pipeline, followed by DTI preprocessing and brain network generation. This undertaking is time-intensive, with each DTI scan necessitating 4-5 hours for preprocessing. Unfortunately, completion won't be immediate. Fortunately, the need to obtain consents again for DTI-derived brain network release isn't necessary. Given our sequence of tackling fMRI preprocessing first, the fMRI collection is now ready for community use, prompting us to release it promptly to enable unimodal studies to commence.
> [No detailed analysis on how parcellation affects model performance] We have updated the following discussions in the 2nd paragraph of Section 5.1: “With respect to parcellation methods, we observe that atlas-based parcellation methods in general outperform clustering-based methods. The best performance on each dataset always occurs when AAL, HO or Schaefer is applied. Particularly, AAL achieves the best performance in 4 out of 6 datasets. The inferiority of clustering-based parcellation is likely due to the effects from some randomness in the clustering process, e.g., initial centroid selection in k-means.”
>
> **[Diverse graph analytics tasks beyond classification]** Please refer to the general response above.
> [Collection size still limited for effective training and evaluation of deeper GNN models] Our release of the collection of brain networks is a beginning. We aim at updating our data collection in a periodic manner to keep up with the data sources of longitudinal nature, e.g., ADNI and PPMI, and new data sources emerging in the future with the data version control implemented via Figshare. This will benefit general study in brain network analysis and perhaps reach the mass required by deeper GNN models in future.
>
> **[Lack of comparisons to other existing neuroimaging datasets]** We have positioned our work by introducing the preprocessed images and brain networks available in the community in the “Data Collection” part of Section 2 - Related Work. In particular, we have discussed PCP on the release of preprocessed images and the issues with its preprocessing pipelines. We have also discussed a few brain network datasets with their limitations on data characteristics (e.g., binarized networks) and sample sizes (e.g., 80 samples).
>
> **[Ablation studies to evaluate the impact of key design choices]** The whole pipeline can be broadly divided into two parts: (A) preprocessing raw neuroimages to produce preprocessed images; and (B) extracting brain networks from preprocessed images.
> The design choice made in part (A) lies in the selection of fMRIPrep. Please refer to the general response for this choice.
> The design choice made in part (B) is on the parcellation schemes. Instead of selecting a single parcellation scheme, Alan and Haribalan proposed to use a set of commonly used parcellation schemes, covering representative schemes in both atlas-based and clustering-based categories. This choice broadens the potential use of this collection by providing more types of brain networks.
>
> We believe with the help of our two domain experts, we have the guarantee of the correctness of the pipeline and the domain-sensible design choices.
>
> **[Lack of summary on limitations and potential future work]** Limitations and potential future work were discussed in the “Conclusions, Limitations, and Future Directions” section of our submission.

---

> ### Comment · Reviewer_ebHo · 2023-08-29
>
> Thanks for sharing the responses which have addressed my concerns

---

### Official Review · Reviewer_NQuQ · 2023-07-21
**This paper presents a collection of preprocessed brain networks datasets for machine learning applications. The datasets were initially preprocessed with unified preprocessing framework and then evaluated with a few Ml methods**

**Rating:** 4
**Confidence:** 4
**Correctness:** 1 - The current benchmarks have been …
**Clarity:** The presentation of the paper could b…

**Strengths:**

This paper addresses the pressing need of collecting brain network datasets, which is a significant topic of study.
The current study considers a sufficient number of datasets from multiple sources.
Five different parcellation methods have been analyzed in the study.


**Additional Feedback:**

-

**Documentation:**

No official documentation is provided by some details are provided on the website

**Ethics:**

Not sure

**Limitations:**


1 - In Section 5.2, the analysis on graph-based datasets is unclear. Specifically, it is not clear how the contrast dense subgraph has been constructed. While I understand there may be space limitations, I believe providing this information is crucial for understanding the dataset construction.

2 - Table 1 mentions that the number of node features is the length of BOLD signals. However, the current paper lacks any information about graph-based datasets and node features. Was the BOLD signals used as node features? Clarification on this point would be helpful.

**Opportunities For Improvement:**


1 - Currently, the paper considers a smaller number of ROIs for constructing the functional connectome. Although fewer ROIs capture the high-level overview of the brain, they ignore the granularity of the brain. I believe the paper should explore using the maximum number of ROIs or at least provide a comparison. Moreover, from an ML perspective, recent studies [1] have shown that a larger number of ROIs lead to improved performance.

2 - I was wondering why the correlation matrices are flattened for training models. Since they are symmetric, I think using only one part of the matrix would suffice. This would also reduce the size of the data by half.

3 - Several fMRI preprocessing pipelines exist, and there is no consensus on which one leads to better performance. The paper should consider a few preprocessing pipelines and provide a comparison. It would also be interesting to show which fMRI preprocessing pipeline leads to improved performance and explain why.








[1] . Said, A., Bayrak, R. G., Derr, T., Shabbir, M., Moyer, D., Chang, C., & Koutsoukos, X. (2023). NeuroGraph: Benchmarks for Graph Machine Learning in Brain Connectomics. arXiv preprint arXiv:2306.06202.

**Relation To Prior Work:**

Yes

**Summary And Contributions:**

This paper presents a collection of preprocessed brain network datasets collected from various sources for machine learning applications. The data is initially preprocessed using a unified fMRI preprocessing method with five different parcellation methods, and the resulting functional connectome has been released as datasets. For evaluation, five conventional machine learning methods and one graph-based method have been trained on flattened correlation matrices, and the accuracy has been reported.

---

> ### Author Response · Authors · 2023-08-22
> **Response to Reviewer NQuQ [1/2]**
>
> **[Comparison with larger number of ROIs]** Thank you for the suggestion. We test different number of ROIs on the Schaefer parcellation from {100, 200, 500, 1000}. The results on ABIDE and TaoWu are presented in the tables below, with the best result of each method highlighted in bold and the best result in each dataset underlined. Our baseline methods perform the best when the number of ROIs is set at 100 or 200; [1] on the other hand, was conducted on a different set of tasks and datasets and thus may get different observations. We have added these results in Section 5.4 of the revision (Page 10).
>
> Note that generating a brain network with user-defined number of ROIs is easy with our released pipeline and the preprocessed images. Researchers have the flexibility of directly feeding the preprocessed images to Step D of our pipeline (Parcellation and ROI Definition) by using different parcellation schemes other than the five used in our paper or setting different number of ROIs in some parcellation schemes (e.g., Schaefer, k-means, and Ward). Subsequent Step E will then generate brain networks with the customized number of ROIs. The most time-consuming step in our pipeline is Step C – fMRIPrep Preprocessing, which requires 4-5 hours per subject. With the availability of preprocessed images generated by fMRIPrep, re-running Steps E and F on different number of ROIs is very instant and convenient. This shows the merits of releasing the pipeline, the code base, and preprocessed images, together with the brain networks.
>
> |     ABIDE    |     100           |     200           |     500             |     1000          |
> |--------------|-------------------|-------------------|---------------------|-------------------|
> |     LR       |     $64.8\pm 3.7$    |     $\mathbf{\underline{67.9\pm 3.8}}$    |     $67.4  \pm 3.8$    |     $67.6\pm 4.7$    |
> |     NB       |     $61.6\pm 3.6$    |     $\mathbf{62.4\pm 3.6}$    |     $61.9\pm 3.0$      |     $60.5\pm 3.5$    |
> |     SVC      |     $\mathbf{64.4\pm 5.1}$    |     $63.7\pm 4.4$    |     $62.4\pm 3.8$      |     $63.1\pm 4.0$    |
> |     kNN      |     $\mathbf{59.7\pm 3.5}$    |     $59.5\pm 5.1$    |     $58.0\pm 3.8$      |     $55.5\pm 5.8$    |
> |     RF       |     $\mathbf{62.6\pm 2.6}$    |     $62.4\pm 3.2$    |     $61.6\pm 3.2$      |     $62.1\pm 5.3$    |
>
> |     TaoWu    |     100            |     200            |     500              |     1000           |
> |--------------|--------------------|--------------------|----------------------|--------------------|
> |     LR       |     $\mathbf{\underline{75.0\pm 15.0}}$    |     $57.5\pm 22.5$    |     $52.5\pm 23.6$    |     $52.5\pm 17.5$    |
> |     NB       |     $\mathbf{62.5\pm 23.6}$    |     $\mathbf{62.5\pm 23.1}$    |     $60.0\pm 20.0$    |     $60.0\pm 20.0$    |
> |     SVC      |     $\mathbf{65.0\pm 15.0}$    |     $57.5\pm 22.5$    |     $55.0\pm 24.5$    |     $50.0\pm 19.4$    |
> |     kNN      |     $\mathbf{65.0\pm 16.6}$    |     $52.5\pm 23.6$    |     $50.0\pm 25.0$    |     $50.0\pm 25.0$    |
> |     RF       |     $\mathbf{57.5\pm 25.1}$    |     $55.0\pm 26.9$    |     $52.5\pm 26.1$    |     $45.0\pm 18.7$    |
>
> **[Why correlation matrices are flattened]** We agree that the correlation matrices are symmetric and either upper or lower part is sufficient in terms of information. Though it is a common practice in the field to flatten the entire correlation matrix (e.g., [a]) and we followed the same mechanism, we are open to exploring compression options for data release.
>
> [a] Ahmed EI-Gazzar, etc al. Dynamic Adaptive Spatio-Temporal Graph Convolution for fMRI Modelling. Machine Learning in Clinical Neuroimaging, 2021.
>
> **[Compare with other fMRI preprocessing pipelines to see which leads to better performance]** Please refer to the general response.

---

> ### Author Response · Authors · 2023-08-22
> **Response to Reviewer NQuQ [2/2]**
>
> **[Construction of contrast dense subgraphs]** Sorry for the confusion. The construction of contrast dense subgraphs is not for the dataset construction but is part of the classification methodology proposed in [28]. The approach in [28] first computes two summary connectivity matrices $C$ and $P$, respectively from the Control group and the Patient group. It then extracts the dense contrast subgraphs from $C$ and $D$ and uses them as features for classification. Essentially, a dense contrast subgraph refers to a subset of nodes whose induced subgraph is dense in $C$ and sparse in $P$ (namely contrast), or vice versa. It can be considered an Optimal Quasi-Clique (OQC) problem [b] and solved by the DENSDP approach [c]. We have updated the revision to make this clear in Page 9.
>
> [b] Tsourakakis, Charalampos, et al. "Denser than the densest subgraph: extracting optimal quasi-cliques with quality guarantees." Proceedings of the 19th ACM SIGKDD international conference on Knowledge discovery and data mining. 2013.
>
> [c] Cadena, Jose, Anil Kumar Vullikanti, and Charu C. Aggarwal. "On dense subgraphs in signed network streams." 2016 IEEE 16th International Conference on Data Mining (ICDM). IEEE, 2016.
>
> **[BOLD signals as node features]** Thank you for bringing this to our attention. Our released collection includes a node feature matrix for each subject, in which each row is the BOLD signal for an ROI. We include the BOLD signals as node feature matrices in our data release for data completeness: some analytical methods on brain networks may want to utilize such information directly or further extract features from them. The node features were not used in the conventional ML methods and the graph-based method [28] in our experiments as the purpose there is to assess the quality of brain network construction, i.e., the connectivity matrices. We have included these discussions in Page 7 of our revision.
>
> **[Lack of data loader]** Our released code base has included the data loader. Please refer to Lines 50-89 in the following file: [https://github.com/bna-data-analysis/extract-brain-network/blob/main/NIPS_paper_gridsearch_experiments.py](https://github.com/bna-data-analysis/extract-brain-network/blob/main/NIPS_paper_gridsearch_experiments.py)
>
> **[More graph ML methods as baselines]** Please refer to the general response.

---

> ### Comment · Reviewer_NQuQ · 2023-08-31
>
> I have read the author response and I want to keep my score as is. Thanks.

---

### Official Review · Reviewer_RUkN · 2023-07-23
**A large dataset of preprocessed connectomes from fMRI for clinical conditions predictions**

**Rating:** 7
**Confidence:** 5
**Correctness:** Nothing to report. It's correct AFAIK.
**Clarity:** Very clearly written.

**Strengths:**

- paper clarify
- use of community standards (BIDS, fMRIprep, scikit-learn)

**Additional Feedback:**

- Please consider showing learning curves
- Complement literature review when it comes to actual prediction scores.

typos:
pipline -> pipeline
singlesite -> single site
whakapapa -> Whakapapa

**Documentation:**

Pretty good.

**Ethics:**

Data are public with proper IRBs.

**Limitations:**

- My main concern in on actual novelty. This paper offers good baselines on a not so recent problem for which many methods and applied papers have been published.
- I would encourage the authors to positions their work more clearly with respect to other similar attempts in the neuroimaging community..

**Opportunities For Improvement:**

- I would appreciate to see learning curves ie prediction scores as a function of the training set size showing the relevance of the dataset size offered here. It will fully convince that the paper opens a new opportunity for the field with data at this scale.

**Relation To Prior Work:**

this can surely be improved.

**Summary And Contributions:**

This work proposes a dataset from 6 different sources, covering 4 brain conditions, and consisting of a total of 2,702 subjects. It consists of preprocessed data where raw fMRI signals are converted to connectomes (here graphs). Nodes in the graph are brain regions and edges are weighted by the correlation between BOLD activations. The connectomes are then used in supervised learning setup where the targets are the brain conditions.

The problem is not particularly novel yet the paper is well written and study is executed according to the latest standards in the field (BIDS, figshare with DOI etc.)

---

> ### Author Response · Authors · 2023-08-22
> **Response to Reviewer RUkN**
>
> **[Prediction scores wrt the training set size]** Thank you for the suggestion. We tune the training set size (by tuning k in k-fold cross validation: the smaller the k, the smaller the training set size is) on the two large datasets ABIDE and ADNI and plot out the prediction scores on the test sets in Figure 3 of the revision (Page 10). The results show that the performance of each ML method improves with the increase in the training set size. This confirms the necessity of having datasets at this scale.
>
> **[Novelty]** Please refer to the general response.
>
> **[Better positioning of our work in neuroimaging community]** As our work focuses on releasing the collection of brain networks with the pipeline and the code base, we have positioned our work by introducing the preprocessed images and brain networks available in the community in the “Data Collection” part of Section 2 - Related Work. In particular, we have discussed PCP on the release of preprocessed images and the issues with its preprocessing pipelines. We have also discussed a few brain network datasets with their limitations on data characteristics (e.g., binarized networks) and sample sizes (e.g., 80 samples).
>
> **[Typos]** Thank you for bringing this to our attention. We have corrected them in the revision.

---

> > ### Comment · Reviewer_RUkN · 2023-08-31
> > **Thanks**
> >
> > Thanks for addressing my concerns and I do acknowledge the significant work needed to deliver such a dataset. I've increased my score.

---

### Author Response · Authors · 2023-08-22
**General response [3/3]**

**[More graph ML methods as baselines @NQuQ]** We evaluate our datasets using both MLP and 6 graph-based ML techniques: GCN, GraphSAGE, GIN, GAT, GatedGCN, and BrainNetCNN. The graph ML methods involve an 8:1:1 data split for training, validation, and testing, using 10-fold cross-validation. In earlier experiments with conventional ML methods, we employed a 5-fold CV data split. To ensure fairness in comparison, we have re-run all conventional ML experiments using the same data split as the graph ML methods and have incorporated these updates in the revised version.

The table below reports the results of graph ML methods on Schaefer. The best result of each dataset is in bold, with those underlined indicating superior performance to conventional ML methods. No single graph ML method can consistently dominate across datasets. In 3 out of 6 datasets, the best graph ML method outperforms the best of conventional ML methods, which shows the potential of graph-based models. We have added these results in Section 5.1 of the revision (Page 8).

| |  ABIDE | ADNI | PPMI | Matai | TaoWu | Neurocon |
|--------------------|---------------------|-------------------|--------------------|----------------------|----------------------|----------------------|
|     GCN            |     $61.0\pm 2.8$      |     $61.6\pm 0.6$    |     $54.0\pm 9.1$     |     $\mathbf{\underline{73.3  \pm 20.0}}$    |     $60.0\pm 29.2$      |     $59.0  \pm 20.7$    |
|     GraphSAGE      |     $63.1\pm 3.1$      |     $61.2\pm 1.7$    |     $55.0\pm 12.9$    |     $63.3  \pm 16.3$    |     $60.0\pm 33.9$      |     $68.5  \pm 15.2$    |
|     GIN            |     $57.0  \pm 3.9$    |     $61.9\pm 0.4$    |     $\mathbf{57.9\pm 8.1}$     |     $53.3  \pm 16.3$    |     $65.0\pm 20.0$      |     $68.5  \pm 15.2$    |
|     GAT            |     $60.9\pm 5.0$     |     $61.3\pm 1.3$    |     $55.0\pm 8.0$     |     $68.3  \pm 18.9$    |     $\mathbf{67.5  \pm 22.5}$    |     $54.0\pm 15.6$      |
|     GatedGCN       |     $63.6\pm 4.7$      |     $\mathbf{62.1\pm 4.5}$    |     $52.6\pm 11.5$    |     $58.3  \pm 8.3$     |     $65.0\pm 22.9$      |     $\mathbf{\underline{69.0  \pm 25.5}}$    |
|     BrainNetCNN    |     $\mathbf{\underline{65.8  \pm 2.5}}$    |     $61.1\pm 2.9$    |     $57.3\pm 10.3$    |     $66.7  \pm 21.1$    |     $65.0\pm 27.8$      |     $66.0\pm 22.5$      |


**[Graph analytics tasks beyond classification @ebHo, iqnx]** Thank you for the suggestion. Indeed, we have two multi-class datasets in our collection, ADNI and PPMI, with each subject labelled as healthy or in different stages of AD/PD. Besides classification, they could also be used for ordinal regression. The accuracy of the logistic ordinal regression vs that of the logistic regression (classification) are shown in the tables below. The results show that the logistic ordinal regression does not exhibit superiority to the logistic regression. One potential reason may be its sensitivity to the class imbalance problem: ADNI and PPMI have very skewed class distribution as shown in Table II in the supplementary. With the availability of the datasets, researchers could further inspect this issue and design better ordinal regression solutions to handle them. We have added these results and discussions in Section 5.2 of the revision (Page 9).

| ADNI | AAL  |   HO |  Schaefer | K-means  |  Ward  |
|------------------------------------|---------------------|---------------------|---------------------|---------------------|---------------------|
| Logistic Regression |     $64.1  \pm 1.8$    |  $61.9  \pm 2.1$  |  $62.0  \pm 4.2$  |     $\mathbf{58.6  \pm 2.6}$    |     $\mathbf{60.4  \pm 0.8}$    |
| Logistic Ordinal Regression    |     $\mathbf{65.6\pm 2.2}$      |     $\mathbf{62.1\pm 3.0}$      |     $\mathbf{62.8  \pm 2.2}$  |  $58.5\pm 5.5$  |  $60.2\pm 3.3$      |

|     PPMI  |     AAL  |     HO  |     Schaefer  |     K-means |  Ward   |
|------------------------------------|---------------------|---------------------|---------------------|---------------------|---------------------|
|     Logistic Regression   |     $\mathbf{56.0  \pm 7.8}$    |     $56.0  \pm 9.2$    |     $\mathbf{56.5  \pm 6.8}$    |     $\mathbf{60.3  \pm 7.4}$    |     $60.3  \pm 8.8$    |
|     Logistic Ordinal Regression    |     $54.1\pm 13.0$     |     $\mathbf{57.5\pm 8.1}$      |     $55.6\pm 8.9$      |     $59.4\pm 8.5$      |     $\mathbf{60.8\pm 9.1}$      |

---

### Author Response · Authors · 2023-08-22
**General response [2/3]**

**[Compare with other fMRI preprocessing pipelines to see which leads to better performance @NQuQ; Ablation studies on key design choices @ebHo]** The fMRI preprocessing (Step C of Figure 1) takes raw neuro-images as input and produces preprocessed neuro-images for subsequent brain network construction. This step aims to meet quality objectives in neuroscience, such as signal-to-noise ratio improvement, motion artifact reduction, and accurate image registration within the brain outline. Optimizing image quality should not rely on different objectives, like classification performance, to prevent introducing bias that could compromise image quality guarantees. Given the quality control focus of the preprocessing step, our reliance for selecting the fMRI preprocessing tool rests on domain experts Alan Wang and Haribalan Kumar. Alan Wang is a professor at the University of Auckland, New Zealand, with affiliations to the Centre for Brain Research and the Auckland Bioengineering Institute. Haribalan Kumar is a skilled MRI scanning/preprocessing specialist at General Electric Healthcare (GE), a leading MRI scanner brand. The decision to opt for fMRIPrep was guided by the following factors.

1) Cutting-edge Pipeline: fMRIPrep, an advanced pipeline introduced in Nature Methods (2019), receives 1722 citations and is adopted by neuro-imaging research groups in eminent institutions like Stanford University, University of Melbourne, and Facebook AI Research for neuro-imaging preprocessing.

2) Efficient Automation: fMRIPrep streamlines preprocessing by integrating tools from FSL, ANTs, FreeSurfer, and AFNI. Unlike manual use of these packages, fMRIPrep automates workflows, saving time and effort, and provides visual quality control reports.

3) User-Friendly Robustness: fMRIPrep offers flexibility in preprocessing steps and stands out for its transparency. Unlike alternatives like DPARSF, FreeSurfer, CONN, GraphVar, and Brain Connectivity Toolbox, it's less opaque.

Alan and Hari's iterative evaluation ensures quality output. For instance, the surface reconstruction, not a default fMRIPrep setting, was enabled after their assessment to enhance data alignment across subjects, requiring an additional 3 computing hours per subject in preprocessing overhead.

---

### Author Response · Authors · 2023-08-22
**General response [1/3]**

We thank all the reviewers for their highly constructive comments that help improve the paper quality. We also thank the reviewers for the recognition of our work in addressing the pressing need of brain network datasets (**NQuQ**, **ebHo**, **iqnx**), being clearly and well written (**RUkN**, **v8Nt**), following the latest standards in the field (**RUkN**), covering diversified data sources, clinical conditions and parcellation schemes (**ebHo**), incorporating domain expert input (**ebHo**), the proposed methodology being sound and well described (**v8Nt**).

We respond to some general comments in this thread before providing point-to-point responses. Our revised paper is uploaded to the OpenReview system with the revised parts highlighted in blue.

We hope our rebuttal address the reviewers' questions and concerns. We are more than happy to discuss with all reviewers if they still have any unresolved concerns or additional questions about the paper or our rebuttal.

**[Contribution and novelty @RUkN]** The main novelty we see in our work lies in the curation and release of the large brain network collection. Though many brain network analysis methods have been developed, most of the datasets they used were kept private. Without a publicly accessible large collection of brain networks, the barriers to performing fair comparisons across different methods and attracting more researchers to this interdisciplinary field remain high. This motivated our work on providing ready-to-use and high-quality brain networks to the research community, which to the best of our knowledge, is a contribution to the field.

We have devoted significant manpower, computing resources, and time to come up with a domain-sensible and up-to-date pipeline, to preprocess the raw images, and to negotiate with different parties to reach consent for releasing the derived brain networks to the general public. Such efforts may not have been fully reflected in the paper. For example, to fill the gap of lacking ready-to-use brain networks for Alzheimer’s disease, we have initiated intense communications and discussions with the stakeholders of ADNI, the largest AD data source, during the past year regarding various issues on releasing the derived data from ADNI under its strict data policy, leading to their unprecedented decision to establish a process for external data submissions within the ADNI data system. Another example lies in the release of our Matai dataset. Our team performed the data collection (which requires the rigorous design and execution of user and data agreements, on-site data acquisition, purchase of advanced scanners, involvement of imaging engineers, etc.), the subsequent data processing, as well as getting the consent of data releasing from the Maori advisory board. The inclusion of the Matai dataset further increases the data diversity in terms of clinical conditions.

We believe that our efforts in building the “infrastructure” in this important interdisciplinary area will benefit the research community.

---

### Decision · Program_Chairs · 2023-09-22

**Decision:**

Accept (Poster)

**Comment:**

The reviewers appreciated this contribution as a useful source of preprocessed data and benchmarks. They did note the importance of situating the work with regards to connected-based predictive models benchmarks in the resting-state literature, as well as a more systematic post-hoc study of the impact of pipeline choices.